# *AutoCGP*: Closed-Loop Concept-Guided Policies from Unlabeled Demonstrations

**Pei Zhou**[1,*] **Ruizhe Liu**[1,*] **Qian Luo**[1,5,*] **Fan Wang**[3,4] **Yibing Song**[3,4] **Yanchao Yang**[1,2]

[1]HKU Musketeers Foundation Institute of Data Science, The University of Hong Kong
[2]Department of Electrical and Electronic Engineering, The University of Hong Kong
[3]DAMO Academy, Alibaba Group  [4]Hupan Lab  [5]Transcengram
`{pezhou,zrllrz360,qianluo}@connect.hku.hk`
`{fan.w,songyibing.syb}@alibaba-inc.com,yanchaoy@hku.hk`

## Abstract

Training embodied agents to perform complex robotic tasks presents significant challenges due to the entangled factors of task compositionality, environmental diversity, and dynamic changes. In this work, we introduce a novel imitation learning framework to train closed-loop concept-guided policies that enhance long-horizon task performance by leveraging discovered manipulation concepts. Unlike methods that rely on predefined skills and human-annotated labels, our approach allows agents to autonomously abstract manipulation concepts from their proprioceptive states, thereby alleviating misalignment due to ambiguities in human semantics and environmental complexity. Our framework comprises two primary components: an *Automatic Concept Discovery* module that identifies meaningful and consistent manipulation concepts, and a *Concept-Guided Policy Learning* module that effectively utilizes these manipulation concepts for adaptive task execution, including a *Concept Selection Transformer* for concept-based guidance and a *Concept-Guided Policy* for action prediction with the selected concepts. Experiments demonstrate that our approach significantly outperforms baseline methods across a range of tasks and environments, while showcasing emergent consistency in motion patterns associated with the discovered manipulation concepts. Codes are available at: https://github.com/PeiZhou26/AutoCGP.

## 1 Introduction

The pursuit of developing robotic systems that can perform a wide array of tasks in diverse environments is a core mission in embodied AI and robotics. While imitation learning has been employed to derive policies from extensive robotic datasets (Collaboration et al., 2024; Kim et al., 2024), it remains challenging for embodied agents to effectively learn policies for long-horizon tasks. Errors and inconsistencies can accumulate, making situations progressively more dynamic and complex to manage. Additionally, imitation learning faces further difficulties due to the varying and unpredictable conditions encountered at each step (Brohan et al., 2022; 2023; Padalkar et al., 2023; Fu et al., 2024).

To tackle long-horizon tasks, it is essential to divide them into sequential sub-tasks with specific goals. This strategy offers two main advantages: first, breaking down a complex task into simpler sub-tasks makes the learning process more manageable by focusing on attainable goals, thereby reducing overall complexity and enhancing feasibility. Second, these sub-tasks often involve common actions, such as "grasping," which are frequently repeated in various tasks. By learning these common skills, the training dataset is more effectively utilized, allowing for reuse in new, unseen tasks. Therefore, it is beneficial to learn a representation that captures the intrinsic patterns of these common actions, referred to as *manipulation concepts* (Shao et al., 2021; Liu et al., 2024a; Jia et al., 2024; Zhou & Yang, 2024). These concepts are analogous to human motor skills like grasping, throwing, pushing, and pulling (see Fig. 1). Just as a set of basic skills can facilitate humans to perform complicated tasks such as cooking or assembling furniture, the learned manipulation concepts can also be composed or reused to address complex long-horizon robotic manipulation tasks.

---

*: First Authors

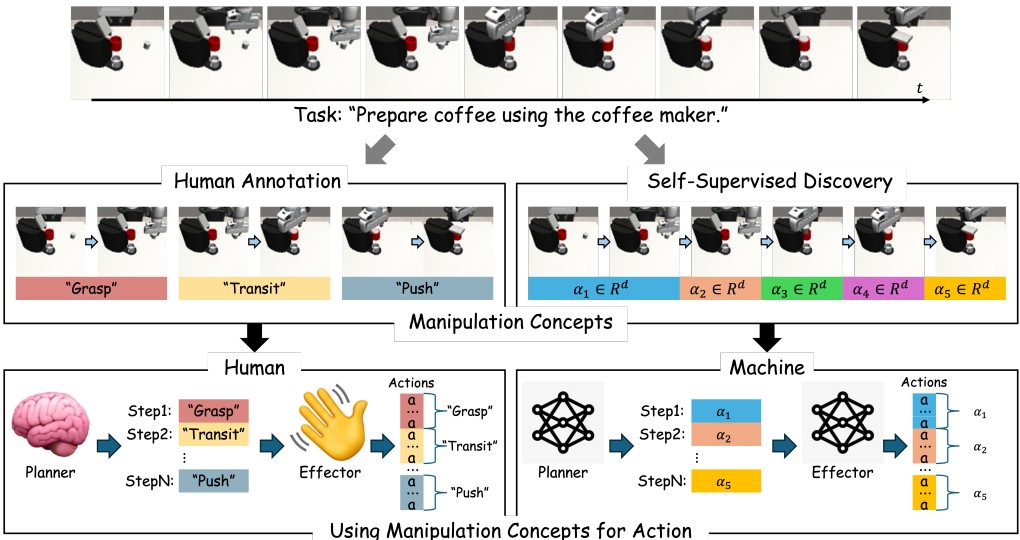

Figure 1: Comparison of human defined manipulation concepts with those derived by our method. Human manipulation concepts, or motor skills, can be described in natural language and combined for long-horizon tasks. In contrast, our method applies a self-supervised learning framework to discover meaningful manipulation concepts as feature vectors (see Sec. 3.1). These manipulation concepts can be applied to manipulation tasks via composition. For humans, manipulation concepts inform planning, which guides specific actions. Similarly, our model execute policies guided by the discovered manipulation concepts (see Sec. 3.2).

Many prior works have relied on human semantics, e.g., using natural language, to label sub-processes and enhance imitation learning by training hierarchical policies. These approaches often depend on manually crafted annotations, heuristic rules, or large models trained for encoding human prior (Mo et al., 2019; Ahn et al., 2022; Eisner et al., 2022; Li et al., 2023c; Di Palo et al., 2023; Ding et al., 2023; Lin et al., 2023). This reliance poses significant challenges: the extensive manual effort required for data collection is time-consuming and resource-intensive, and the inherent subjectivity may lead to manipulation concepts that do not align well with the robot's configuration or its operational environment, ultimately hindering effective learning and performance.

To address these challenges of manual annotation, task complexity, and environmental unpredictability, we propose a novel framework that combines automatic concept discovery with closed-loop concept-guided policy learning. Our approach autonomously extracts and utilizes manipulation concepts directly from the robot's proprioceptive states (Please refer to Sec. A.1 for the rationale behind using proprioceptive states), thereby eliminating the dependency on predefined skills and human-annotated labels. This self-supervised method not only mitigates the misalignment caused by ambiguities in human semantics but also adapts dynamically to unforeseen situations.

Specifically, our framework consists of two main components. The first is the *Automatic Concept Discovery* module, which enables robots to derive manipulation concepts in a bottom-up manner without resorting to human annotators. This module focuses on key characteristics shared by a wide range of skills and sub-goals, allowing for the abstraction of high-quality and consistent manipulation concepts across various tasks. The second component is the *Closed-Loop Concept-Guided Policy Learning* module. This module establishes an adaptive policy by employing a *Concept Selection Transformer (CST)* to propose and adjust manipulation concepts in real-time during the robot's interactions with the environment. This dynamic selection enables the system to adapt to environmental changes and select the most appropriate concept for the current situation. Subsequently, the *Concept-Guided Policy (CGP)* utilizes the selected manipulation concepts to predict actions based on instantaneous visual inputs, leveraging a diffusion policy to generate efficient and high-dimensional actions. This closed-loop approach ensures that the policy can dynamically adjust and refine its actions based on continuous feedback, effectively addressing the challenges of dynamic and unpredictable environments.

The experimental results demonstrate that our pipeline significantly improves manipulation task performance, surpassing baseline methods across a variety of tasks and environments. We observe

emergent consistency within these manipulation concepts, particularly in terms of motion patterns. Our contributions can be categorized into three major aspects: **1**) A novel closed-loop concept-guided learning pipeline that discovers and utilizes manipulation concepts from unlabeled demonstrations; **2**) A manipulation concept discovery method that abstracts high-quality, consistent manipulation concepts in various tasks, reducing dependence on human prior knowledge; and **3**) A concept-guided policy that incorporates learned manipulation concepts with a diffusion policy, achieving state-of-the-art performance on various benchmarks.

## 2 RELATED WORK

**Concept discovery in manipulation tasks**. The manipulation concepts we propose are discrete symbolic representations, akin to natural language and codes, which are widely used and beneficial for learning manipulation tasks. Despite advances in improving manipulation tasks through costly manual labeling (Mo et al., 2019; Eisner et al., 2022; Li et al., 2023c), unsupervised concept discovery remains under-explored. Recent advances in Large Language Models (LLMs) show promise for automated labeling (Ahn et al., 2022; Di Palo et al., 2023; Ding et al., 2023; Lin et al., 2023), but they lack sufficient grounding in real-world sensor data (Zhou & Yang, 2024). Studies adopting self-supervised methods to discover manipulation concepts (Sermanet et al., 2018; Yan et al., 2020; Zambelli et al., 2021; Morgan et al., 2021; von Hartz et al., 2022; Weng et al., 2023; Liu et al., 2024a) include utilizing information-theoretic tools such as mutual information (Gregor et al., 2016; Hausman et al., 2018; Hu et al., 2024) and modeling temporal relationships and latent features using approaches like time-contrastive learning (Nair et al., 2022; Ma et al., 2023), identifying critical temporal junctures (Jayaraman et al., 2018; Neitz et al., 2018; Pertsch et al., 2020; Zhu et al., 2022; Caldarelli et al., 2022), and predicting future time-steps (Starke et al., 2022; Li et al., 2024). Many of these methods also integrate geometric (Morgan et al., 2021; Zhu et al., 2022; Shi et al., 2023) and physical (Yan et al., 2020) constraints. Several approaches struggle to generalize due to data inefficiency, heuristic designs and reliance on predefined primitives tailored to specific tasks (Wang et al., 2024), while others rely on overly broad self-supervised principles that lack focus on the intended concept use cases. Recent methods like BeT (Shafiullah et al., 2022) and VQ-BeT (Lee et al., 2024) utilize self-supervised learning to derive discrete symbols from actions. However, their focus remains on mitigating continuous noise in neural network outputs through action discretization rather than on capturing higher-level abstractions, such as motor skills. The proposed discovery mechanism efficiently manages generalization across robots with the same morphology while capturing key features of diverse manipulation skills beyond the original action encoding.

**Manipulation policy learning**. A significant body of research is devoted to learning manipulation policies through methods like imitation learning (Argall et al., 2009; Rahmatizadeh et al., 2018; Zhang et al., 2018; Fang et al., 2019) and reinforcement learning. These approaches primarily use deep neural networks to map states to actions, enabling interaction. Key techniques in this field include the Decision Transformer (Chen et al., 2021; Team et al., 2023; Xu et al., 2023b; Zhao et al., 2023; Tanaka et al., 2024) and Diffusion Policy (Wang et al., 2022; Chi et al., 2023; Li et al., 2023b; Pearce et al., 2023; Liu et al., 2024b; Tan et al., 2024; Yan et al., 2024; Ze et al., 2024). To address more complex tasks, recent advancements have integrated self-supervised skill extraction (Peng et al., 2022; Li et al., 2023a) and hierarchical planning with concepts (Xu et al., 2018; Yang et al., 2022; Hutsebaut-Buysse et al., 2022; Jia et al., 2023; Liang et al., 2023) into policy learning, or a combination of them (Peng et al., 2022; Wan et al., 2024). Various methods have also been proposed to handle multiple manipulation concepts within a single model (Brohan et al., 2022; 2023; Driess et al., 2023). Additionally, Large Language Models (LLMs) are leveraged for the massive prior knowledge encoded to manage diverse situations in manipulation tasks (Huang et al., 2023a;b; Long et al., 2023; Wang et al., 2023; Wong et al., 2023; Xie et al., 2023; Yu et al., 2023; Izzo et al., 2024; Zhou et al., 2024). However, these methods often struggle with real-world tasks due to the lack of physical grounding. Our approach overcomes this by deriving manipulation concepts directly from robot observations and proprioceptive states, automatically ensuring physical grounding in an end-to-end manner.

## 3 METHOD

We aim to develop closed-loop concept-guided policies, which output action by taking in a manipulation concept (e.g., indicating a motor skill) and the current state. The manipulation concept in effect

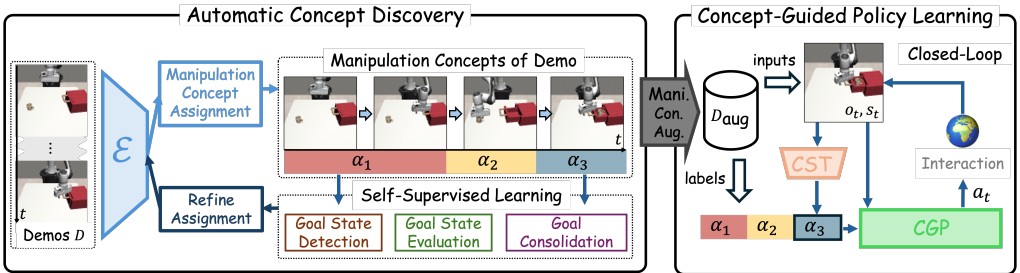

Figure 2: The proposed pipeline for Learning **Closed-Loop Concept-Guided Policies** from unlabeled demonstrations. The *Automatic Concept Discovery* module labels demonstrations in $D$ with manipulation concepts using three strategies (see Sec. 3.1 and Fig. 3). These strategies form a self-supervised learning process, refining the assignment of manipulation concepts with $\mathcal{E}$. The labeled demonstrations form $D_{\text{aug}}$, providing observations ($o_t$ and $s_t$) and labels ($\alpha_k$) for *Concept-Guided Policy Learning*, which trains the Closed-Loop Policy comprising CST and CGP (see Sec. 3.2). In this setup, CST selects the appropriate manipulation concept ($\alpha_k$) to guide CGP in generating actions.

is adjusted according to the task and the current situation (i.e., closed-loop). Moreover, these manipulation concepts are discovered in a self-supervised manner without resorting to human annotations. Specifically, our pipeline consists of two key modules as shown in Fig. 2. The first module, *Automatic Concept Discovery*, detects segments of similar motion patterns from unannotated demonstrations to derive manipulation concepts. The second module, *Concept-Guided Policy Learning*, leverages these manipulation concepts to train a policy that adapts to environmental changes by selecting the most appropriate manipulation concept (*Concept Selection Transformer* (CST)) and predicting actions with the proposed manipulation concept (*Concept-Guided Policy* (CGP)).

## 3.1 AUTOMATIC CONCEPT DISCOVERY

Given unannotated demonstrations $D$, where $\tau \in D$ is a trajectory, i.e., $\tau = \{(s_t^\tau, o_t^\tau, a_t^\tau)\}_{t=1}^{T(\tau)}$ (with $o_t^\tau$ the environmental observation, $s_t^\tau$ the agent's proprioceptive state, $a_t^\tau$ the action at time-step $t$, and $T(\tau)$ the length of $\tau$), the *Automatic Concept Discovery* (ACD) module aims to detect manipulation concepts by assigning each state in a trajectory to one of the $K$ learnable embeddings $\mathcal{A} = \{\alpha_k\}_{k=1}^K$. We denote $\mathcal{K} = \{k\}_{k=1}^K$ as the index set for the manipulation concepts. Specifically, to perform the assignment, we instantiate an encoder $\mathcal{E}$, such that for every $\tau \in D$, $\mathcal{E}$ maps its state at $t$ to an index $\mathbf{k}_t^\tau \in \mathcal{K}$ corresponding to the embedding $\boldsymbol{\alpha}_t^\tau$. Let $\Theta_\mathcal{E}$ be the parameters of $\mathcal{E}$ (excluding $\mathcal{K}$ and $\mathcal{A}$), then we have:

$$\mathbf{k}_t^\tau = \mathcal{E}(t|\tau; \mathcal{K}, \mathcal{A}, \Theta_\mathcal{E}), \quad \boldsymbol{\alpha}_t^\tau = \alpha_{\mathbf{k}_t^\tau}. \tag{1}$$

Note that sub-sequences sharing an index $k \in \mathcal{K}$ belong to the same manipulation concept represented by $\alpha_k$, depicting the behaviors initiated by it. Additionally, the assignment encoder $\mathcal{E}$ prioritizes the proprioceptive information to facilitate deriving coherent manipulation concepts, since proprioceptive motion sequences exhibit high consistency across tasks and different data types.

**The structure of** $\mathcal{E}$ (Eq. 1) is adapted from the VQ-VAE architecture (Van Den Oord et al., 2017), which selects an index from $\mathcal{K} = \{k\}_{k=1}^K$ as the manipulation concept for time step $t$ in a demonstration $\tau$. This selection matches the feature vector generated by the encoder to the nearest embeddings in $\mathcal{A} = \{\alpha_k\}_{k=1}^K$:

$$\mathbf{k}_t^\tau = \mathcal{E}(t|\tau; \mathcal{K}, \mathcal{A}, \Theta_\mathcal{E}) = \arg\min_{k \in \mathcal{K}} \|\tilde{\mathcal{E}}(t|\tau; \Theta_\mathcal{E}) - \alpha_k\|, \quad \boldsymbol{\alpha}_t^\tau = \alpha_{\mathbf{k}_t^\tau}. \tag{2}$$

Here, $\tilde{\mathcal{E}}$ is the sub-module that takes the demonstration $\tau \in D$ as input and outputs a feature of the same dimension as the approximation of the embedding vectors in $\mathcal{A}$ at each time step $t$. The discrete nature of the VQ-VAE encoder, as discussed in the following sections, naturally enables the segmentation of sub-trajectories that achieve sub-goals represented by the same manipulation concepts, facilitating the identification of the moment when the sub-goal is accomplished during training. Please refer to Sec. A.2 for a detailed description on the implementation of $\mathcal{E}$, especially for the preservation of gradients.

**Goal State Detection.** Given the current state, a manipulation concept or sub-goal enables the envisioning of the future state upon its accomplishment. To encourage the discovery of meaningful

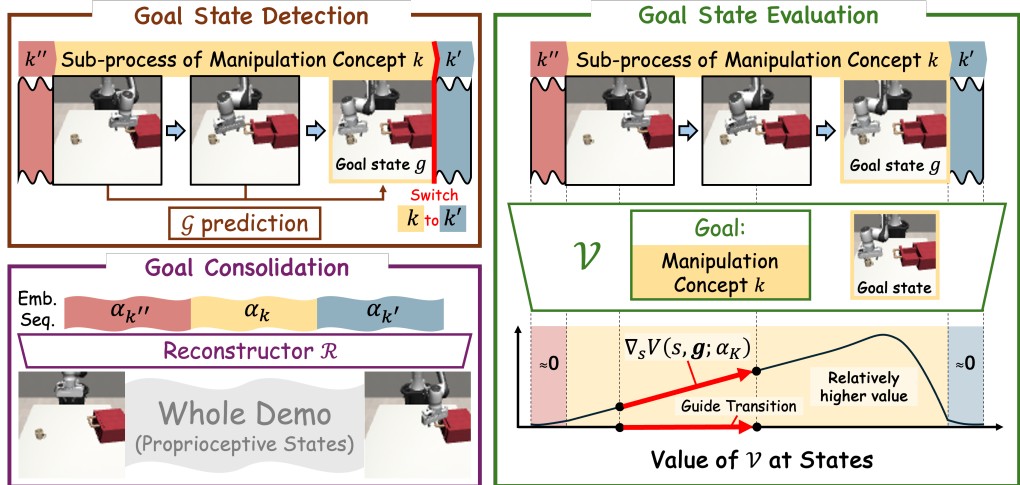

Figure 3: *Automatic Concept Discovery* Strategies. The Automatic Concept Discovery (ACD) module employs three key strategies. The **Goal State Detection** strategy maps each state, along with its manipulation concept (embedding), to the goal state of its sub-process, defined at the junction between segments with different manipulation concepts ($k$ and $k'$, see Eq. 4). The **Goal State Evaluation** strategy uses the value function $\mathcal{V}$ to evaluate both the *Alignment* and the *Completion* of a proprioceptive state based on the manipulation concept and goal state. *Alignment* is indicated by the value difference between states within and outside specific manipulation concept segments. *Completion* is derived from the gradient of $\mathcal{V}$, guiding state transitions. The **Goal Consolidation** strategy maps manipulation concept embeddings (i.e., Emb.) back to their proprioceptive states.

manipulation concepts, we leverage the above phenomenon and design a **Goal State Detection** strategy, which learns by training a network $\mathcal{G}$ that maps each state of $\tau$ with the putatively associated manipulation concept embedding to the goal state (ending state) of the corresponding sub-sequence. This allows manipulation concept embeddings ($\mathcal{A}$) and the assignment encoder ($\mathcal{E}$) to be refined through back-propagation. The objective is the loss between the goal states predicted by $\mathcal{G}$ and those defined by the partitioning of the manipulation concepts:

$$\mathcal{L}^{\text{gd}} = \|\mathcal{G}(\{s_t^\tau, \boldsymbol{\alpha}_t^\tau\}_{t=1}^{T(\tau)}) - (g_t^\tau)_{t=1}^{T(\tau)}\| . \tag{3}$$

Here $\boldsymbol{\alpha}_t^\tau$ is computed according to Eq. 2, and $g_t^\tau$ denotes the proprioceptive state at which the sub-goal for time-step $t$ in $\tau$ is achieved. Leveraging the discrete embeddings in $\mathcal{E}$, once indices from $\mathcal{K}$ are assigned as manipulation concepts to the states in demonstration $\tau$, the state at the sub-goal's accomplishment can be detected by retrieving the nearest time step where the manipulation concept switches, indicating the completion of the current sub-goal:

$$g_t^\tau = s_{\arg\min_u \{u : u \geq t, \mathbf{k}_{u+1}^\tau \neq \mathbf{k}_u^\tau\}}^\tau . \tag{4}$$

Please refer to Fig. 3 for an illustration of the *Goal State Detection* procedure.

**Goal State Evaluation.** Given a state depicting when the sub-goal indicated by a manipulation concept is achieved, one can assess whether a state is well-aligned towards achieving the sub-goal, and can further evaluate the progress toward its *completion* at the current state. To further enhance the quality of the discovered manipulation concepts, we develop a **Goal State Evaluation** strategy. This involves using the assigned manipulation concepts to train a value function $\mathcal{V}$ that evaluates both the *alignment* and the *completion* degrees for a proprioceptive state conditioning on the specific manipulation concept and its associated goal state:

$$\mathcal{V}(s_t^\tau, g_u^\tau; \alpha_k) \to [0, 1], \text{where } \alpha_k \in \mathcal{A} . \tag{5}$$

A detailed implementation of Eq. 5 can be found in Sec. A.3, and $\mathcal{V}$, in theory, shall differentiate the compatibilities of a state against various manipulation concepts by their values. States compatible with the manipulation concept $\alpha_k$ that aiming for the goal state $g_u^\tau$ should be assigned higher values, while others with lower values, due to either a mismatch of the manipulation concept or not resulting

in the goal state (Goal State Evaluation in Fig. 3). This discriminative ability (for alignment) of $\mathcal{V}$ could be acquired by training with the following binary classification objective[1]:

$$\mathcal{L}_a^{\text{ge}}(s_t^\tau, g_u^\tau; k) = \begin{cases} -\log \mathcal{V}(s_t^\tau, g_u^\tau; \alpha_k) & \text{if } \mathbf{k}_t^\tau = k \text{ and } g_u^\tau = g_t^\tau, \\ -\log(1 - \mathcal{V}(s_t^\tau, g_u^\tau; \alpha_k)) & \text{otherwise}. \end{cases}$$

$$\mathcal{L}_a^{\text{ge}} = \Sigma_{u=1}^{T(\tau)} \left( \mathbb{I}\{\mathbf{k}_u^\tau \neq \mathbf{k}_{u+1}^\tau\} \cdot \Sigma_{t=1}^{T(\tau)} \mathcal{L}_a^{\text{ge}}(s_t^\tau, g_u^\tau; \mathbf{k}_u^\tau) \right).$$

(6)

However, the compatibility learned according to the above may not faithfully assess the completion degree of a state regarding the sub-goal state, as Eq. 6 does not explicitly enforce that a higher score represents greater progress towards the sub-goal state, thus the ambiguities in the evaluation.

To address this issue, we further constrain the gradient field of $\mathcal{V}$ to enforce accurate *completeness* evaluation for more informative manipulation concepts. We propose that the progress toward completion should be correlated with the increase or decrease in $\mathcal{V}$'s value, which can be effectively captured through its gradients. Moreover, when $g_t^\tau$ and $\alpha_k$ are fixed in Eq. 5, the gradient field with respect to the current state $s_t^\tau$, while $\alpha_k$ is in effect, should guide the state transition, represented by $\nabla s_t^\tau$ (Goal State Evaluation in Fig. 3). Thus, we train a network $\Pi$, which maps the current state and its gradient to this state transition as follows:

$$\mathcal{L}_c^{\text{ge}} = \Sigma_{t=1}^{T(\tau)} \left\| \Pi\left(s_t^\tau, \nabla_{s_t^\tau} \mathcal{V}(s_t^\tau, g_t^\tau; \boldsymbol{\alpha}_t^\tau)\right) - \nabla s_t^\tau \right\|, \tag{7}$$

where the state transition $\nabla s_t^\tau$ can be computed as the discrepancy between the states from two consecutive time steps, i.e., finite difference.

**Goal Consolidation.** Furthermore, we propose that all states associated with a given manipulation concept shall exhibit similarity to some extent, as they share the same sub-goal or skill. To utilize this characteristic, we introduce a reconstruction network $\mathcal{R}$ trained to map the embedding vector back to its corresponding state:

$$\mathcal{L}^{\text{gc}} = \left\| s_{1:T(\tau)}^\tau - \mathcal{R}\left(\boldsymbol{\alpha}_{1:T(\tau)}^\tau\right) \right\|. \tag{8}$$

We denote this as a consolidation aimed at promoting the consistency of sub-sequences assigned with the same manipulation concept, ensuring that similar states serve similar purposes.

Together, Eq. 9 constitute the training objective for the self-supervised discovery of manipulation concepts from unannotated demonstrations:

$$\mathcal{L}^{\text{ACD}} = \mathcal{L}^{\text{gd}} + \mathcal{L}_c^{\text{ge}} + \lambda_{ent}\left(\mathcal{L}_a^{\text{ge}} + \mathcal{L}^{\text{vq}}\right) + \lambda_{gc}\mathcal{L}^{\text{gc}}. \tag{9}$$

Here $\mathcal{L}^{\text{vq}}$ is the combination of the commitment loss and vector quantization loss in Van Den Oord et al. (2017), and $\lambda_{ent}$ and $\lambda_{gc}$ are positive hyperparameters. Next, we describe the learning of closed-loop concept-guided policies with the discovered concepts.

## 3.2 CLOSED-LOOP CONCEPT-GUIDED POLICY LEARNING

With the self-supervised discovered manipulation concepts, we augment the unannotated demonstrations $D = \{\tau = \{(s_t^\tau, o_t^\tau, a_t^\tau)\}_{t=1}^{T(\tau)}\}$ by tagging each state in a demonstration with a manipulation concept $\mathbf{k}_t^\tau \in \mathcal{K}$, resulting in the augmented dataset $D_{\text{aug}} = \{\tau = \{(s_t^\tau, o_t^\tau, a_t^\tau, \mathbf{k}_t^\tau)\}_{t=1}^{T(\tau)}\}$. With $D_{\text{aug}}$, we can utilize the discovered and grounded manipulation concepts $\mathcal{K}$ to learn a closed-loop visuomotor policy:

$$k \sim p_{\text{CST}}(k|o, s; \Theta_{\text{CST}})$$
$$a \sim \pi_{\mathcal{D}}(a|o, s, k; \Theta_{\text{CGP}})$$

(10)

where $\Theta_{\text{CST}}$ are the parameters of $p_{\text{CST}}$ and $\Theta_{\text{CGP}}$ are the parameters of $\pi_{\mathcal{D}}$. The proposed framework enables the policy to dynamically select and adjust the manipulation concept in response to environmental changes with the *Concept Selection Transformer* (CST): $k \sim p_{\text{CST}}(k|o, s; \Theta_{\text{CST}})$, and to execute appropriate actions based on real-time visual inputs and corresponding manipulation concepts through the *Concept-Guided Policy* (CGP): $a \sim \pi_{\mathcal{D}}(a|o, s, k; \Theta_{\text{CGP}})$ in the following.

---

[1]The loss $\mathcal{L}_a^{\text{ge}}$ in Eq. 6 focuses on the time step when the manipulation concept ID switches, similar to Eq. 4. Once the switch is determined, the binary classification objective is applied to every time step of the trajectory.

**Concept Selection Transformer (CST).** The *Concept Selection Transformer* (CST) determines the manipulation concept based on run-time observations: $\mathbf{k}_t \sim p_{\text{CST}}(k|o_t, s_t)$, where $o_t$ and $s_t$ denote the observation and proprioceptive state, respectively, and $\mathbf{k}_t$ is the selected manipulation concept from $\mathcal{K}$. This design allows adjustments at any time step, enabling the system to adapt to environmental changes and select the most appropriate concept for the current situation. The training leverages cross-entropy loss to align the predicted probability distribution over manipulation concepts with the one-hot labels corresponding to the manipulation concepts provided by $\mathcal{E}$:

$$\mathcal{L}_{\text{CST}} = -\mathbb{E}_{\tau \sim D, t \sim U(0, T(\tau))} \log p_{\text{CST}}(k = \mathbf{k}_t^\tau | o_t^\tau, s_t^\tau; \Theta_{\text{CST}}). \tag{11}$$

**Concept-Guided Policy (CGP).** The *Concept-Guided Policy* (CGP) $\pi_{\mathcal{D}}$ utilizes manipulation concepts predicted by $p_{\text{CST}}$ within an imitation learning framework. We adopt the Diffusion Policy (Chi et al., 2023) for its ability to generate complex, high-dimensional actions from learned distributions. Given varying stages and sub-goals in a manipulation process, CGP adjusts itself with respect to the conditional manipulation concept provided by the CST module. Since the sub-trajectories labeled with a specific manipulation concept are always shorter than the original $\tau$, the learning of the skills indicated by these manipulation concepts is usually more efficient due to reduced complexity and can lead to improved performance due to compositional generalization.

Specifically, we leverage a diffusion policy $\pi_{\mathcal{D}}$ conditioned on the observed environment, the robot's proprioceptive state, and the selected manipulation concept. It then denoises the action from a random sample drawn from $\mathcal{N}(0, I)$ using a DDPM framework with the denoising module $\mathcal{D}$ (Eq. 13 and Eq. 14). As stated in Chi et al. (2023), this approach yields actions that are more robust to multiple viable options and dynamic environments compared to other methods. To enhance the coupling between $p_{\text{CST}}$ and $\pi_{\mathcal{D}}$, we utilize the entire manipulation concept distribution $\mathcal{T}(o_t, s_t; \Theta_{\text{CST}})$ predicted by $p_{\text{CST}}$ rather than simply selecting the manipulation concept with the highest probability:

$$a_t \sim \pi_{\mathcal{D}}(a|o_t, s_t, \mathcal{T}(o_t, s_t; \Theta_{\text{CST}}); \Theta_{\text{CGP}}),$$
$$\text{where } \mathcal{T}(o_t, s_t; \Theta_{\text{CST}}) = [p_{\text{CST}}(k|o_t, s_t; \Theta_{\text{CST}})]_{k=1}^K \in \mathbb{R}^K. \tag{12}$$

This enables the joint training of $p_{\text{CST}}$ and $\pi_{\mathcal{D}}$, allowing $p_{\text{CST}}$ to refine its manipulation concept (distribution) prediction with not only the augmented manipulation concepts from *Automatic Concept Discovery*, but also with the feedback from training $\pi_{\mathcal{D}}$ using the action prediction loss.

More precisely, for the DDPM training of the denoising module $\mathcal{D}$ with demonstrations $\tau \in D_{\text{aug}}$, we minimize the prediction loss between the injected random noise $\epsilon^{(n)}$ (for time step $n$) and the predicted noise from the noisy action $e_t^{\tau,(n)}$, where $a_t^\tau$ is perturbed by $\epsilon^{(n)}$. The DDPM loss is jointly optimized with the loss for the *Concept Selection Transformer* in Eq. 11 to enhance $p_{\text{CST}}$'s selection of manipulation concepts and its coordinated behavior with $\pi_{\mathcal{D}}$:

$$\mathcal{L}_{\text{CGP}} = \mathbb{E}\left[\left\|\mathcal{D}(\epsilon_t^{\tau,(n)}|o_t^\tau, s_t^\tau, \mathcal{T}(o_t^\tau, s_t^\tau; \Theta_{\text{CST}}); \Theta_{\text{CGP}}) - \epsilon^{(n)}\right\|^2\right],$$
$$\mathcal{L}_{\text{policy}} = \mathcal{L}_{\text{CGP}} + \lambda \mathcal{L}_{\text{CST}}. \tag{13}$$

Here, $\mathcal{L}_{\text{CST}}$ is the loss for training $p_{\text{CST}}$ (Eq. 11), and $\lambda > 0$ is the hyper-parameter for weighting. For action inference, it incrementally refines the generated actions, beginning with random Gaussian noise, by applying a reverse diffusion process, resulting in $a_t^{(1)}$ as the sampled (predicted) action:

$$\pi_{\mathcal{D}} : a_t^{(n-1)} = \beta_n \left(a_t^{(n)} - \gamma_n \mathcal{D}(\epsilon_t^{(n)}|s_t, o_t, \mathcal{T}(o_t, s_t; \Theta_{\text{CST}}); \Theta_{\text{CGP}})\right) + \sigma_n \mathcal{N}(0, I). \tag{14}$$

Leveraging the diffusion model's strong capability to model complex distributions, $\pi_{\mathcal{D}}$ can produce actions of higher quality when conditioned on the selected manipulation concept. For the detailed architecture and training pipeline, please refer to Sec. B.

## 4 EXPERIMENTS

We evaluate the proposed pipeline for *Automatic Concept Discovery* and *Closed-Loop Concept-Guided Policy* learning on various manipulation tasks. Our primary evaluation compares the performance of our method with major baseline approaches across diverse tasks within the Robosuite simulation environment (Zhu et al., 2020). Moreover, we conduct an ablation study to analyze the

Table 1: Success rates of policies trained with the proposed method and competing baselines across diverse manipulation tasks. DP represents the standard diffusion policy, while all other methods with "+DP" build on this framework by incorporating various concept discovery approaches.

| | Cof. | | | Ham. | | Stk. 3 | | 3 Pc. | | | Thd. | | Mug. | |
|---|---|---|---|---|---|---|---|---|---|---|---|---|---|---|
| | $D_0$ | $D_1$ | $D_2$ | $D_0$ | $D_1$ | $D_0$ | $D_1$ | $D_0$ | $D_1$ | $D_2$ | $D_0$ | $D_1$ | $D_0$ | $D_1$ |
| BeT | 0.66 | 0.52 | 0.42 | 0.88 | 0.36 | 0.60 | 0.32 | 0.42 | 0.14 | 0.00 | 0.34 | 0.18 | 0.34 | 0.26 |
| DP | 0.80 | 0.58 | 0.40 | **1.00** | 0.46 | 0.68 | 0.68 | 0.54 | 0.28 | 0.00 | 0.60 | **0.28** | 0.76 | 0.28 |
| InfoCon+DP | 0.90 | 0.70 | 0.58 | 0.96 | 0.58 | 0.62 | 0.52 | 0.66 | 0.22 | 0.02 | 0.72 | 0.16 | 0.78 | 0.36 |
| Xskill+DP | 0.88 | 0.74 | 0.50 | **1.00** | 0.54 | 0.72 | **0.76** | 0.74 | 0.36 | 0.00 | 0.70 | 0.20 | 0.70 | 0.40 |
| AWE+DP | 0.86 | 0.68 | 0.48 | 0.96 | 0.56 | 0.70 | 0.68 | 0.60 | 0.26 | 0.00 | 0.62 | 0.16 | 0.76 | 0.30 |
| Ours+DP | **0.98** | **0.84** | **0.72** | **1.00** | **0.66** | **0.78** | 0.72 | **0.82** | **0.42** | **0.04** | **0.80** | **0.28** | **0.88** | **0.50** |

contributions of the different components within our manipulation concept discovery and policy learning modules. We demonstrate the advantages of the discovered manipulation concepts and validate the efficacy of the conditional diffusion policy in improving task performance under the closed-loop framework.

**Implementation details.** We evaluate our method on tabletop manipulation tasks as described in MimicGen (Mandlekar et al., 2023), detailed in Sec. C.1. These tasks are categorized into six types: Coffee Making (Cof.), Hammer Cleanup (Ham.), Stack Three Cubes (Stk. 3), Three Piece Assembly (3 Pc.), Threading (Thd.), and Mug Cleanup (Mug.). Each task is associated with several levels of environmental variability and noise, denoted as $D_0$, $D_1$, and $D_2$, where a larger subscript indicates greater variability and higher noise levels. For each task and its corresponding level of variation, we select 950 demonstrations provided by MimicGen for Imitation Learning. The training samples include front-view images, wrist-view images, robot proprioceptive state, and ground-truth actions at each timestep. We follow the procedures described in Sec. 3 to train our concept-guided policies (detailed in Sec. B): **1) Automatic Concept Discovery:** First, we apply our framework's *Automatic Concept Discovery* method across six task types, using only the 950 demonstrations with the highest level of variation for each task (e.g., $D_2$ for Cof.). Task type information is omitted in this process. We train on mixed demonstrations from all tasks, and then label all demonstrations and their respective levels of initial variation with the encoder $\mathcal{E}$ in Eq. 1. **2) CST and CGP:** For each task, we use a *Concept Selection Transformer* (CST) to map inputs—comprising front-view images, wrist-view images, and robot state—to the relevant execution concepts. Concurrently, a *Concept-Guided Policy* (CGP) mentioned in Sec. 3.2 is trained to map the CST inputs and manipulation concepts into actions. These two processes are jointly trained to enhance synergy and performance.

**Baselines.** We first train the baseline **Diffusion Policy (DP)** without leveraging manipulation concepts. We then compare our method with other major concept discovery baselines (implementation details in Sec. A.5). Here is a brief summary: **1) InfoCon** (Liu et al., 2024a): InfoCon is a self-supervised framework that discovers manipulation concepts through generative and discriminative informativeness regarding the low-level physical state and state changes. **2) XSkill** (Xu et al., 2023a): XSkill is an imitation learning framework that leverages self-contrastive learning to extract and transfer manipulation skills from unlabeled human and robot videos. In this case, we reimplement XSkill using only robotic demonstrations. **3) AWE** (Shi et al., 2023): AWE enhances robotic imitation learning by heuristically extracting minimal waypoints from demonstrations, streamlining the decision-making process by approximating entire trajectories through linear interpolation. We also include **BeT** (Shafiullah et al., 2022) as a baseline since it employs learnable discrete representations similar to manipulation concepts.

## 4.1 QUANTITATIVE RESULTS

**Evaluation.** Our primary evaluation metric is the task success rate (SR), which varies based on the specific criteria for each task (see Sec. C.1 for details). For instance, in the Coffee Making task, success is defined as accurately grasping the coffee and placing it into the coffee machine, while in the Mug Cleanup task, success involves moving the mug from the table into a drawer. The observation space for all tasks includes two images (each $112 \times 112$) from the front and wrist views, along with 9-dimensional proprioceptive information of the agent (gripper position, rotation, and opening state). During evaluations, the environment is initialized randomly. We conduct tests over 50 episodes

Table 2: Ablation study of our proposed *Automatic Concept Discovery* module. We show the success rates of ablations of different components in the module.

|  | Cof. | | | Ham. | | Stk. 3 | | 3 Pc. | | Thd. | Mug. | |
|---|---|---|---|---|---|---|---|---|---|---|---|---|
|  | $D_0$ | $D_1$ | $D_2$ | $D_0$ | $D_1$ | $D_0$ | $D_1$ | $D_0$ | $D_1$ | $D_0$ | $D_0$ | $D_1$ |
| w/o GD | 0.92 | 0.80 | 0.58 | 0.98 | 0.58 | 0.70 | 0.68 | 0.72 | 0.38 | 0.72 | **0.88** | 0.48 |
| w/o GE | 0.92 | 0.78 | 0.58 | **1.00** | 0.64 | 0.72 | 0.64 | 0.80 | 0.40 | 0.78 | 0.78 | 0.40 |
| w/o GC | 0.96 | **0.84** | 0.60 | 0.94 | 0.56 | 0.68 | 0.68 | 0.74 | 0.38 | **0.80** | 0.72 | 0.46 |
| Ours | **0.98** | **0.84** | **0.72** | **1.00** | **0.66** | **0.78** | **0.72** | **0.82** | **0.42** | **0.80** | **0.88** | **0.50** |

for each task using a set of fixed random seeds for fair comparison. Given the complexity of these multi-step manipulation tasks, we set a maximum of 600 rollout steps per episode for evaluation.

**Main results.** Tab. 1 presents the task success rates (SR) of the trained policies across 50 randomly initialized environments, with variations in object positions and placement angles. We use SR to evaluate the quality of the discovered concepts, since a higher success rate indicates that the manipulation concepts learned by the model can guide the policy towards task completion more effectively. The results demonstrate that our proposed method consistently outperforms DP baseline and other concept discovery pipelines on the majority of tasks. In particular, for relatively challenging tasks like the Cof.(D2) task and the Mug.(D1) task, the performance improvement is even more pronounced, indicating that our method excels in handling more complex, long-horizon tasks. This suggests that our approach is highly effective at discovering intrinsically meaningful manipulation concepts that enhance the learned policies' ability to generalize across dynamic environments during robotic execution. The discovery of these meaningful concepts highlights the practical benefits of the proposed closed-loop framework in solving manipulation tasks with high variability and complexity.

**Ablation study.** We conduct an ablation study to assess the impact of various components within our concept discovery and policy learning pipeline, with results detailed in Tab. 2. As described in Sec. 3.1, the *Automatic Concept Discovery* module consists of three primary components. This ablation involves systematically disabling each component to evaluate their individual contributions to concept discovery and subsequent policy training. The results, as shown in Tab. 2, demonstrate that omitting any of these components adversely affects performance across tasks. Specific settings tested include disabling goal state detection (w/o GD), goal state evaluation (w/o GE), and goal consolidation (w/o GC). The table illustrates that policies trained with a fully integrated concept discovery module achieve the best performance across a variety of tasks, underscoring the importance of each component in enhancing policy performance with improved manipulation concept quality.

## 4.2 VISUAL RESULTS

We also visualize and assess the concepts discovered by our method. First, we demonstrate the consistency of the concepts identified across different manipulation trajectories of the same task (Coffee D2), as illustrated in Fig. 4 (upper). The results highlight how well our method generalizes the identified concepts within variations of the same task. We also compare our results with those from XSkill (Xu et al., 2023a), shown in Fig. 4 (lower). For XSkill embeddings, we utilize K-means clustering to group the embeddings into separate concepts, allowing for a direct comparison between the two methods. This comparison underscores a key advantage of our approach: our method consistently identifies similar, robust manipulation concepts across different trajectories of the same task, even when there are variations during execution. This suggests that our framework effectively captures the underlying motion pattern of the task. In contrast, the concepts identified by the XSkill pipeline exhibit greater variability and are prone to noise. Furthermore, Fig. 5 illustrates that sub-sequences corresponding to the same manipulation concepts exhibit similarity in proprioceptive motion, even when these sub-sequences are derived from different tasks. This suggests that the discovered concepts capture underlying commonalities in dynamics, regardless of the task context. These similarities highlight the generalizability and robustness of the concepts, making them transferable across various tasks. Please see Sec. C.3 for more visualizations.

## 5 DISCUSSION

We develop a closed-loop concept-guided manipulation policy learning pipeline with two modules: *Automatic Concept Discovery* and *Concept-Guided Policy Learning*, which abstracts manipulation concepts from unannotated robot demonstrations and uses them to improve manipulation policy

efficacy in an automatic manner. Our findings highlight the significance of using autonomously discovered manipulation concepts to enhance policy training in complex manipulation tasks. However, a key limitation is the predefined maximum number of concepts due to the codebook structure. Additionally, this pipeline has not yet been tested on more complex morphology, e.g., humanoid. Future research could focus on improving generalization across diverse real-world scenarios and developing automated methods to assess concept consistency, which is further discussed in Sec. D.

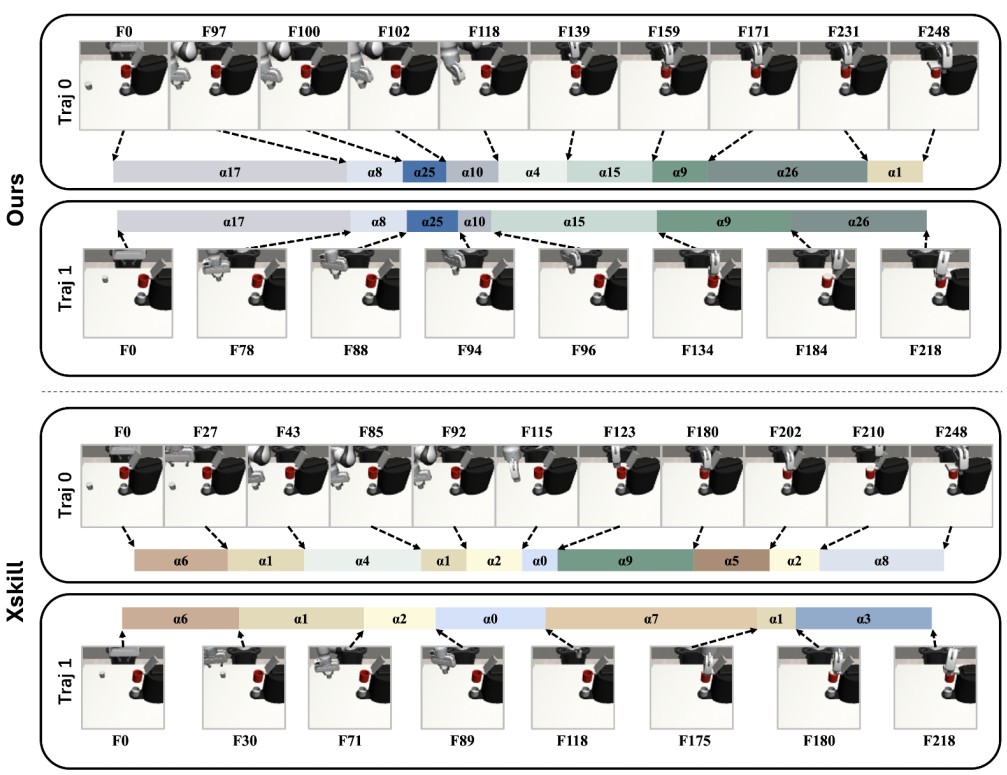

Figure 4: Visual comparison of manipulation concepts discovered by our method versus Xskill (Xu et al., 2023a). F{number} denotes the frame index of an image within a trajectory. Our method demonstrates significantly greater consistency across different trajectories compared to Xskill.

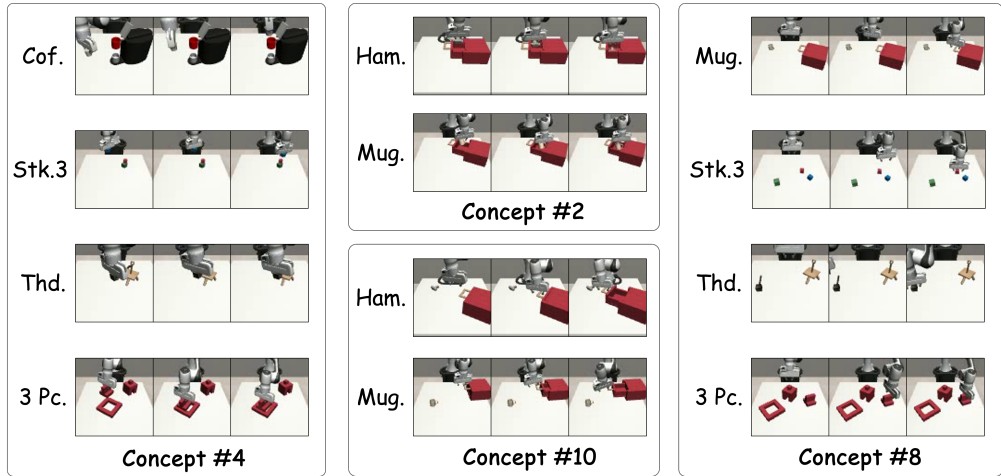

Figure 5: Consistency check of the concepts discovered by our method across different tasks. The sub-sequences under the same concept from different tasks share similarities in motion. Concept #4: robot arms are placing the objects in target locations. Concept #8: robot arms are moving toward the objects and preparing to pick. Concept #10: robot arms are pulling the drawer handles.

**Acknowledgement:** This work is supported by the Early Career Scheme of the Research Grants Council (grant # 27207224), the HKU-100 Award, a donation from the Musketeers Foundation, and DAMO Academy through the Alibaba Innovative Research Program.

**Ethics Statement:** Our research explores concepts that may go beyond common semantics. For now, we are concentrating on extracting concepts from basic manipulation tasks and will maintain this focus. While the algorithm holds potential for advancing toward more complex concepts, we intend to proceed cautiously, carefully considering ethical implications and the need for responsible control mechanisms.

**Reproducibility Statement:** We provide our implementation details of our Manipulation Concept Discovery (including baselines) and Concept-Guided policies in the appendix (Sec. A and Sec. B) and the supplementary material. Our experiments use the environments based on simulator Robosuite (Zhu et al., 2020). We welcome discussion and advice on better and more stable performance of our pipeline and further implementation improvement in more practical environments.

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

## A  CONCEPT DISCOVERY IMPLEMENTATION

### A.1  WHY WE USE PROPRIOCEPTION DATA

Our decision to utilize proprioceptive state data is guided by two considerations:

***Representation of Inherent Motion Patterns.*** Manipulation concepts are analogous to human motor skills as they primarily encapsulate proprioceptive movements. In human cognition, we broadly categorize actions without specifying the object involved—for instance, the action is generalized as "grasping" rather than distinguishing between "grasping an apple," "grasping cloth," or "grasping a phone." This categorization emphasizes the movements of our hands rather than the objects being grasped, underscoring a strong linkage between manipulation concepts and proprioceptive states. Utilizing proprioceptive states as a foundation for learning these manipulation concepts provides a direct pathway to acquiring representations that are deeply integrated into the robot's natural motion dynamics. This method aligns with prior research (Shafiullah et al., 2022; Lee et al., 2024), which utilized action sequences to convert changes in proprioceptive states into discretized representations, capturing the dynamics at individual time steps.

***Consistency Across Contexts.*** By focusing on proprioceptive states, we inherently filter out the contextual variability associated with different tasks and environmental setups. This methodological choice ensures greater consistency in the manipulation concepts learned, regardless of task-specific differences. For instance, the manipulation concept of "grasping" might be employed by a robot across various scenarios involving distinct objects. Proprioceptive sequences during these tasks are more likely to remain consistent than sequences derived from environmental data, which may fluctuate with changes in object types, spatial arrangements, and background settings. Such consistency underscores the potential for developing manipulation concepts that are not only more stable but also broadly transferable across different operational contexts.

We conducted an additional experiment by integrating environmental observations (images) into the concept discovery process across several tasks. The table below presents the comparative results:

Table 3: Success rates for policies trained using manipulation concepts from two different settings. **Ours** refers to our method leveraging proprioceptive states to discover manipulation concepts. **Img** refers to the approach using environmental camera images.

|  | Cof. $D_2$ | Mug. $D_1$ |
|---|---|---|
| **Ours** | **0.72** | **0.50** |
| Img | 0.64 | 0.40 |

The results indicate a decrease in performance when manipulation concepts are derived from images compared to those derived from proprioceptive states. Nevertheless, we acknowledge the importance of visual information in improving our discovery of manipulation concepts. Moving forward, we aim to conduct further research on effectively optimizing the use of visual inputs. Please refer to Sec. D for additional details.

### A.2  IMPLEMENTATION OF $\mathcal{E}$

We use VQ-VAE Encoder Van Den Oord et al. (2017) as the basic structure of $\mathcal{E}$ in Eq. 1, since it can output a feature vector from an index-able code-book and preserve the gradient flow from continuous encoder $\tilde{\mathcal{E}}$ (Eq. 2):

$$
\begin{aligned}
z_t^\tau &= \tilde{\mathcal{E}}(t|\tau; \Theta_\mathcal{E}) \\
\mathbf{k}_t^\tau &= \arg\min_{k \in \mathcal{K}} \|z_t^\tau - \alpha_k\|, \\
\boldsymbol{\alpha}_t^\tau &= \alpha_{\mathbf{k}_t^\tau}. \\
\boldsymbol{\alpha}_t^\tau &= \mathrm{SG}(\boldsymbol{\alpha}_t^\tau - z_t^\tau) + z_t^\tau
\end{aligned}
\tag{15}
$$

Here we implement $\tilde{\mathcal{E}}$ as a transformer that outputs a feature vector for each time-step $t$ of $\tau \in D$, $\mathrm{SG}(\cdot)$ is the stop gradient operator. The fourth line in Eq. 15 makes sure that the gradient in $\boldsymbol{\alpha}_t^\tau$ can be propagated back to $z_t^\tau$.

### A.3 IMPLEMENTATION OF VALUE FUNCTION $\mathcal{V}$

For implementation of Eq. 5, We utilize a Hypernetwork to strike a better balance between data adaptability and parameter efficiency (Chauhan et al., 2024). This deep neural network takes in proprioceptive states and outputs a feature vector.

$$\mathrm{HN}_{\alpha_k}(s_t^\tau), \quad \alpha_k \in \mathcal{A} \tag{16}$$

Here $\mathrm{HN}_*(\cdot)$ is a hyper-network, and its parameters are generated from the input of concept $\alpha_k \in \mathcal{A}$. We use the Sigmoid-processed cosine similarity to form the value function in Eq. 5. Here $\langle \cdot, \cdot \rangle$ represents the scalar product of two vectors:

$$\mathcal{V}(s_t^\tau, g_u^\tau; \alpha_k) = \mathrm{Sigmoid}\left(T \cdot \frac{\langle \mathrm{HN}_{\alpha_k}(s_t^\tau), \mathrm{HN}_{\alpha_k}(g_u^\tau) \rangle}{\|\mathrm{HN}_{\alpha_k}(s_t^\tau)\|_2 \cdot \|\mathrm{HN}_{\alpha_k}(g_u^\tau)\|_2}\right) \tag{17}$$

Here $T > 0$ is the hyper-parameter, we set it as 5.0 to make full use of the values Sigmoid can have ($\frac{1}{1+e^{-5.0}} \approx 0.9933$).

### A.4 PSEUDO-CODES

We present the pseudocode for the *Automatic Concept Discovery* pipeline at Alg. 1, designed to derive manipulation concepts from demonstrations $\tau \in D$. The provided example assumes a training batch containing a single $\tau$ from $D$, which is analogous to the case when the batch contains multiple $\tau$.

---

**Algorithm 1** Automatic Concept Discovery

**Input**:      Demo $D = \{\tau = (s_t^\tau, o_t^\tau, a_t^\tau,)_{t=1}^{T(\tau)}\}$
**Modules**:   $\tilde{\mathcal{E}}, \mathcal{A} = \{\alpha_k\}_{k=1}^K$ and index $\mathcal{K} = \{k\}_{k=1}^K$
             Goal State Detection: $\mathcal{G}$
             Goal State Evaluation: $\mathcal{V}(\mathrm{HN}_*)$, $\Pi$,
             Goal Consolidation: $\mathcal{R}$
**Output**:     trained state encoder $\phi$

---

**for** iteration of training **do**
     sample $\tau \sim D$
     **for** $t = 1, 2, ..., T(\tau)$ **do**
         $z_t^\tau = \tilde{\mathcal{E}}(t|\tau; \Theta_\mathcal{E})$
         $\mathbf{k}_t^\tau = \arg\min_{k \in \mathcal{K}} \|z_t^\tau - \alpha_k\|$                 ▷ VQ-VAE Encoder Select Manipulation Concepts
         $\boldsymbol{\alpha}_t^\tau = \alpha_{\mathbf{k}_t^\tau}$
         $\boldsymbol{\alpha}_t^\tau = \mathrm{SG}(\boldsymbol{\alpha}_t^\tau - z_t^\tau) + z_t^\tau$                                 ▷ Preserve Gradient
     **end for**
     **for** $t = 1, 2, ..., T(\tau)$ **do**
         Calculate $g_t^\tau$ using Eq. 4
     **end for**
     Calculate $\mathcal{L}^{\mathrm{gd}}, \mathcal{L}_a^{\mathrm{ge}}, \mathcal{L}_c^{\mathrm{ge}}, \mathcal{L}^{\mathrm{gc}}$                      ▷ See Eq. 3, Eq. 6, Eq. 7, Eq. 8
     Calculate $\mathcal{L}^{\mathrm{vq}}$     ▷ VQ loss and Commitment loss according to (Van Den Oord et al., 2017)
     $\mathcal{L}^{\mathrm{ACD}} = \mathcal{L}^{\mathrm{gd}} + \mathcal{L}_c^{\mathrm{ge}} + \lambda_{ent}(\mathcal{L}_a^{\mathrm{ge}} + \mathcal{L}^{\mathrm{vq}}) + \lambda_{gc}\mathcal{L}^{\mathrm{gc}}$
     Back propagation from $\mathcal{L}^{\mathrm{ACD}}$
**end for**

---

### A.5 IMPLEMENTATION OF BASELINES

- **Our method.** All the transformers used in our Concept Discovery Module refer to the structure of transformers used in (Brown et al., 2020) and have an inner embedding feature of 128 dimensions with 8 heads. The network $\mathcal{E}$ in Eq. 1 contains $\tilde{\mathcal{E}}$ (Eq. 2), which is a 4-layer transformer, and a VQ-VAE of 30 codebook items as $\mathcal{A}$. The model $\mathcal{G}$ in Eq. 3 is a 2-layer transformer. The hyper-network $\mathrm{HN}_*$ in Eq. 16 is able to generate a feed-forward linear network of 2 hidden layers to former $\mathcal{V}$ in 5. The $\Pi$ used in Eq. 7 is a 1-layer transformer. The $\mathcal{R}$ used in Eq. 8 is a 4-layer transformer. We employ the AdamW optimizer, coupled with a warm-up cosine annealing scheduler to modulate

the learning rate. This scheduler initiates at 0.1 times the base learning rate, linearly increases the rate to the base level over the course of 1000 epochs, and subsequently reduces the learning rate to 0.1 times the base rate following a cosine function. The weight decay is always $1.0 \times 10^{-3}$. We append all input sequences to the length of $440$ and use a batch size of 16 during training. We train our model for 4000 epochs with a base learning rate of $1.0 \times 10^{-4}$. The loss term Eq. 3 and Eq. 7 receive a weight of 1.0. The loss term Eq. 6 and Eq. 8 receive a weight of $1.0 \times 10^{-3}$. The training process can be finished on a single GeForce RTX 3090 in 1.5 days.

- **XSkill.** Based on the design of Xskill Xu et al. (2023a), we implement the skill discovery framework on the Mimicgen dataset, which only contains the "robot" embodiment in the Xskill pipeline. We use the default hyperparameters as in the Xskill code base. For the temporal skill encoder, we use a 3-layer CNN network followed by an MLP layer as the vision backbone. Same as Xu et al. (2023a), we augment the images in the input video clip by a randomly selected operation from a set of image transformations, including random resize crop, color jitter, grayscale, and Gaussian blur. For the Skill Alignment Transformer (SAT), we also use a standard ResNet-18 as our state encoder. The transformer encoder is composed of 16 layers, each layer featuring a transformer encoder with 4 attention heads. Additionally, the feedforward network within each layer has a dimensionality of 512. We set the training batch size to 28 and a learning rate of $1.0 \times 10^{-4}$. The training takes 4 days to converge on a GeForce RTX 4090 GPU.

- **InfoCon.** Based on the design of InfoCon, we refer to the structure of the transformer used in (Brown et al., 2020). All the size of hidden features output by transformers and concept features is 128 here[2]. The state encoder and state reconstructor both use a 4-layer transformer. The goal-based policy uses a 1-layer transformer. The predictor for the generative goal uses a 2-layer transformer. For hyper-network used for discriminative goals, we use 2 hidden layers in the goal function. The number of concepts is fixed, the maximum number of 30 manipulation concepts for all the tasks. We employ the AdamW optimizer, coupled with a warm-up cosine annealing scheduler same as "**Our method**". The weight decay is always $1.0 \times 10^{-3}$. We append all input sequences to the length of $440$ and use a batch size of 16 during training. We train our model for 4000 epochs with a base learning rate of $1.0 \times 10^{-4}$. Other hyper-parameters are aligned with the work (Liu et al., 2024a). The training process can be finished on a single GeForce RTX 3090 in 1.5 days.

- **AWE.** In the work of AWE, they set thresholds for the end condition of the dynamic programming of finding way-points with different manipulation tasks. Here we modify the method so that it can discover a fixed number (here we choose 10) of key states for all sequences. Since the dynamic programming process naturally discovers a solution with the increasing of a key state number, we can just let it stop at a certain key state number.

## B  CONCEPT-GUIDED POLICY IMPLEMENTATION

### B.1  POLICY ARCHITECTURE

For *Concept Selection Transformer* (CST), we employ a transformer architecture to effectively capture the temporal relationships within a sequence of historical observations and proprioceptive information. This setup is designed to predict the manipulation concepts to be executed, based on current observations. We utilize the standard transformer model, which incorporates a series of self-attention layers and feed-forward networks, as described in (Vaswani et al., 2017). This transformer has 16 standard self-attention blocks with a feedforward dimension of 512. For *Concept-Guided Policy* (CGP), we use the U-Net architecture, a robust CNN-based network. Initially, the front and wrist images are processed through separate ResNet-18 networks to extract image features. These features are then combined with proprioceptive information and the manipulation concepts predicted by CST. This combined data is repeatedly injected into each block of the policy network to enhance the model's responsiveness to conditional information. Thus, the inputs to the CGP include the concatenated features and additional noise. Furthermore, we employ sinusoidal position embedding to embed diffusion timestep information, and within each block, Feature-wise Linear Modulation is used for integrating these features, following the method introduced in (Perez et al., 2018). The architecture of CGP is illustrated in Fig. 6.

---

[2]We use proprioceptive states.

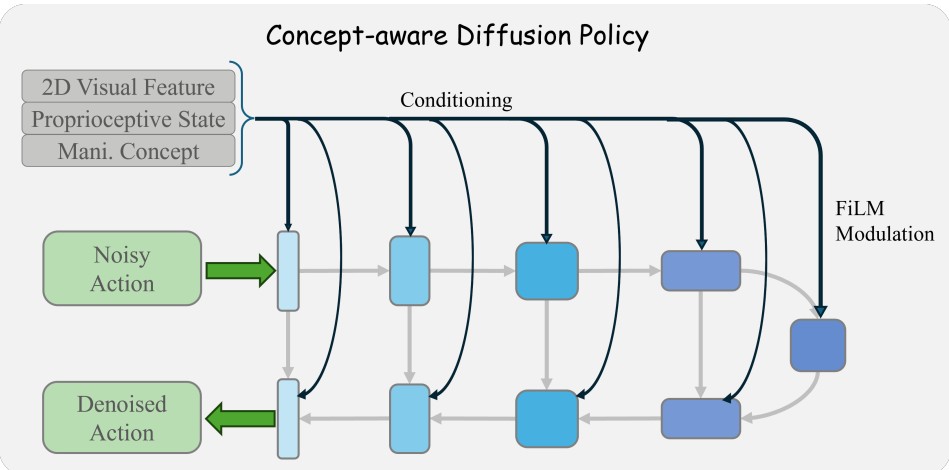

Figure 6: The architecture of concept-aware diffusion policy

## B.2 TRAINING DETAILS

For training, each task (along with its respective levels of variation) utilizes 950 randomly sampled demonstrations (see also Sec. 4). The policy network maintains an input observation length of 4 and an action prediction horizon of 8, with a batch size set at 150. To minimize overfitting, we employ the AdamW optimizer, incorporating a linear warmup of the learning rate. The initial learning rate is set at $1 \times 10^{-4}$ and gradually decreases following a cosine annealing schedule as the number of iterations progresses. Training consists of 100 epochs in total. Additionally, we apply a weight decay of $1 \times 10^{-6}$ to improve model generalization. For diffusion model training, we employ a beta schedule called squaredcos_cap_v2, with a beta range of 1e-4 to 2e-2, which optimizes the generation process by smoothly and controllably adjusting the introduction of noise levels. The CST and CGP are trained simultaneously. The training process can be completed in less than one day on a single GeForce RTX 4090 GPU.

## C MORE ABOUT EXPERIMENTS

### C.1 TASK DETAILS

In our experiment, we utilize the MimicGen dataset for the following six categories of tasks (Fig. 7):
(a) **Coffee**: Insert a cylindrical coffee packet into the coffee machine and secure the lid.
(b) **Hammer Cleanup**: Store the hammer in a drawer.
(c) **Mug Cleanup**: Place the mug into a drawer.
(d) **Stack Three**: Stack three blocks in the following order from top to bottom: blue, red, green.
(e) **Threading**: Thread a needle through a hole.
(f) **Three Piece Assembly**: Assemble three components together.

### C.2 ABOUT ABLATION STUDY

Here, we present an analysis on the granularity of the discovered manipulation concepts. Since each segment labeled with a specific manipulation concept corresponds to a sub-process that achieves the sub-goal associated with that concept, we calculate the average length of these segments across each task presented in Tab. 2 for both the ablation cases and our proposed approach to assess granularity. The results are summarized in Tab. 4. As observed, the manipulation concept segments in the non-ablated version are shorter than those in the ablation cases. This suggests that the performance degradation due to ablation *may* stem from the less fine-grained manipulation concepts. This finding aligns with our hypothesis: shorter segments correspond to simpler sub-tasks, thereby facilitating enhanced policy learning.

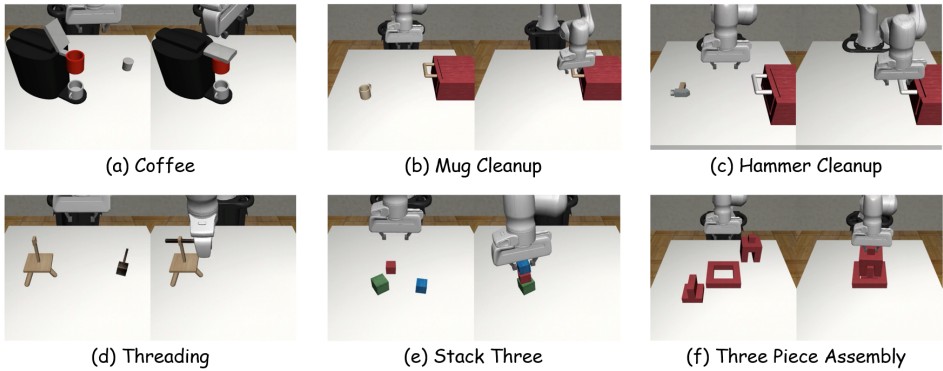

(a) Coffee          (b) Mug Cleanup          (c) Hammer Cleanup

(d) Threading          (e) Stack Three          (f) Three Piece Assembly

Figure 7: Visualization of the 6 categories of tasks in our experiments.

Table 4: Ablation Study of the Proposed *Automatic Concept Discovery* Module: Average segment lengths for ablations of different components within the module, compared to the non-ablated version.

| | Cof. | | | Ham. | | Stk. 3 | | 3 Pc. | | Thd. | Mug. | |
|---|---|---|---|---|---|---|---|---|---|---|---|---|
| | $D_0$ | $D_1$ | $D_2$ | $D_0$ | $D_1$ | $D_0$ | $D_1$ | $D_0$ | $D_1$ | $D_0$ | $D_0$ | $D_1$ |
| w/o GD | 72.0 | 78.3 | 75.3 | 68.5 | 80.5 | 78.7 | 85.7 | 70.4 | 70.0 | 70.3 | 76.3 | 72.2 |
| w/o GE | 54.3 | 59.0 | 44.0 | 52.0 | 53.8 | 57.8 | 51.6 | 58.8 | 58.3 | 52.8 | 61.0 | 51.6 |
| w/o GC | 36.2 | 39.3 | 63.8 | 34.5 | 32.3 | 38.5 | 27.5 | 29.4 | 29.3 | 52.8 | 29.7 | 31.1 |
| Ours | **28.3** | **31.0** | **27.6** | **33.9** | **29.5** | **26.7** | **22.5** | **28.5** | **28.8** | **33.3** | **29.3** | **28.5** |

## C.3 CONCEPT VISUALIZATIONS

In this section, we present visualizations of the manipulation concepts identified through Automatic Concept Discovery (Sec. 3.1) and demonstrate their consistency across various tasks.

Fig. 9 and Fig. 10 illustrate the proprioceptive state patterns associated with different manipulation concepts across different tasks, following a methodology similar to Ye et al. (2024). Specifically, In Fig. 9 and Fig. 10, each row corresponds to a task and each column corresponds to a manipulation concept. The sub-figure at the intersection of a row and a column visualizes all the proprioceptive states from the task in the row that are associated with the manipulation concept in the column (Notably, the analysis begins by applying PCA to the proprioceptive states across all tasks, so the visualized states here are the PCA-reduced representations). The visualization results show that the non-empty sub-figures in each column share similar patterns, highlighting the consistency of the learned manipulation concepts across different tasks.

Please also see Fig. 11, Fig. 12, Fig. 13 and Fig. 14 for visualizations of video frames from sub-processes in different tasks under the same manipulation concepts, demonstrating the consistency of manipulation concepts across different tasks. We also apply the Automatic Concept Discovery process to real-world data from BridgeData V2 (Walke et al., 2023), demonstrating its ability to discover manipulation concepts that exhibit consistency across diverse scenarios. The concepts are visualized in Fig. 15.

Since MimicGen exhibits different levels of variability, as introduced in the **Implementation details** section of Sec. 4, we also observe that demonstrations from the same task can involve relatively distinct sub-processes. Specifically, we find that different sub-processes are often associated with distinct manipulation concepts. For example, in Fig. 8, we illustrate two sub-processes from separate demonstrations of the task **threading**. Both sub-processes occur in the phase "after grasping the needle, transitioning it to align with the hole." However, due to differences in initial positions, one demonstration requires rotating the needle approximately 180 degrees, while the other involves only a slight rotation. These variations lead to the assignment of different manipulation concepts to the respective sub-processes.

Figure 8: We observe an interesting phenomenon: for the same task, if sub-processes differ, they are assigned different sets of manipulation concepts. In this case, both sub-processes involve the transition from grasping the needle to aligning it with the hole. However, in trajectory example 1, the needle must be rotated approximately 180 degrees, while in trajectory example 2, only a minor rotation is required. For the first sub-process, three manipulation concepts are assigned in sequence: #17 → #1 → #26. In contrast, the second sub-process is assigned a single concept, #15.

## D  FUTURE WORKS

***Beyond Proprioception.*** We posit that a promising method to further capitalize on whole environmental observations (e.g., images of the entire environment) is through a "multi-modality" approach. Our future research will concentrate on modeling the interrelations among various modalities, leveraging diverse sensor data from robots to enhance the Automatic Concept Discovery process. However, we face limitations with the datasets currently available; most open-source datasets for robotics primarily include vision and proprioceptive states, along with some linguistic descriptions, but often lack other modalities (such as sound). Moving forward, we will treat proprioceptive states and images of environmental observations as distinct "modalities" and apply multi-modal learning techniques to enhance the Automatic Concept Discovery process. We plan to expand our approach to include additional modalities as we collect our own data or gain access to more comprehensive datasets.

***Persuasive Analysis on Manipulation Concept.*** Future work will also focus on conducting a persuasive analysis of manipulation concepts. While task success rates are a common metric in robotics research, they may introduce unfairness due to variations in how different policy types align with specific manipulation concepts. Some sets of manipulation concepts may be better suited to particular types of policies, while others align more effectively with different policies. This underscores the need for a more **rigorous and insightful investigation into manipulation concepts themselves**, as well as a deeper examination of the relationship between manipulation concepts and policies. Such an exploration is critical for developing a more reasonable and coherent integration of the two. These efforts aim to build a stronger foundation for understanding manipulation concepts and enhancing their integration with policies.

***Potential of learning-from-play (Lynch et al., 2020).*** We would like to emphasize the similarity between the learning-from-play settings and our settings. Our method shares key characteristics with a learning-from-play scenario, as it does not depend on task descriptions or explicit task objectives (Please refer to the Implementation details in Sec. 4), such as natural language descriptions of tasks. This approach essentially mirrors the collection of an unannotated dataset comprising several demonstrations. In this setup, the Automatic Concept Discovery process operates without assuming any knowledge of the specific goals being pursued in any given demonstration sequence. This is strongly analogous to the learning-from-play setting, where the robot observes sequences or demonstrations that are not tied to explicit objectives (Lynch et al., 2020). This resemblance highlights the relevance and adaptability of our method to contexts similar to learning-from-play. Moving forward, we plan to explore the application of our method directly within a learning-from-play scenario.

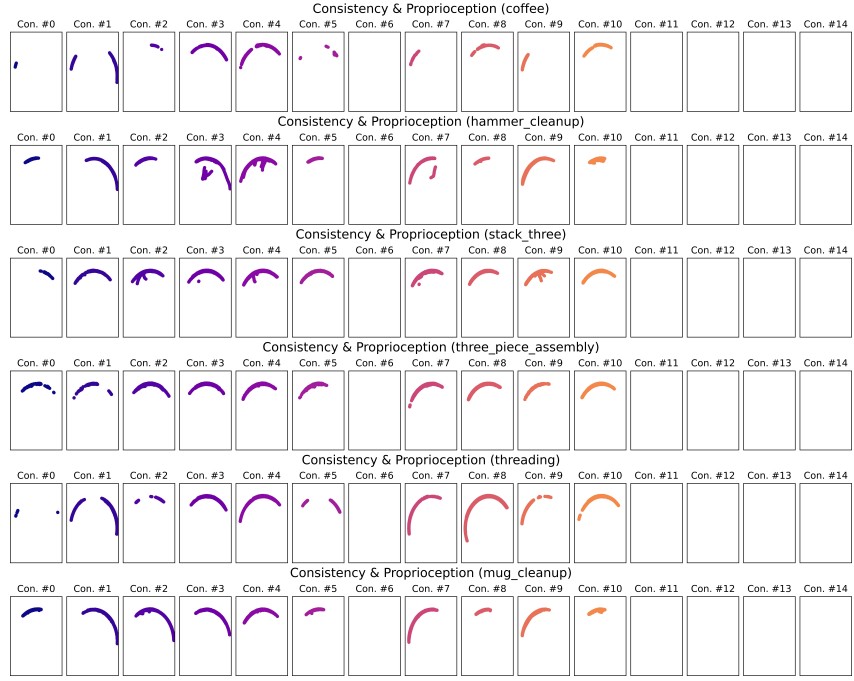

Figure 9: Consistency Visualization (part 1, manipulation concepts #0~14). Each row corresponds to a task and each column corresponds to a manipulation concept. The sub-figure at the intersection of a row and a column visualizes the PCA-reduced proprioceptive states from the task in the row that are associated with the manipulation concept in the column.

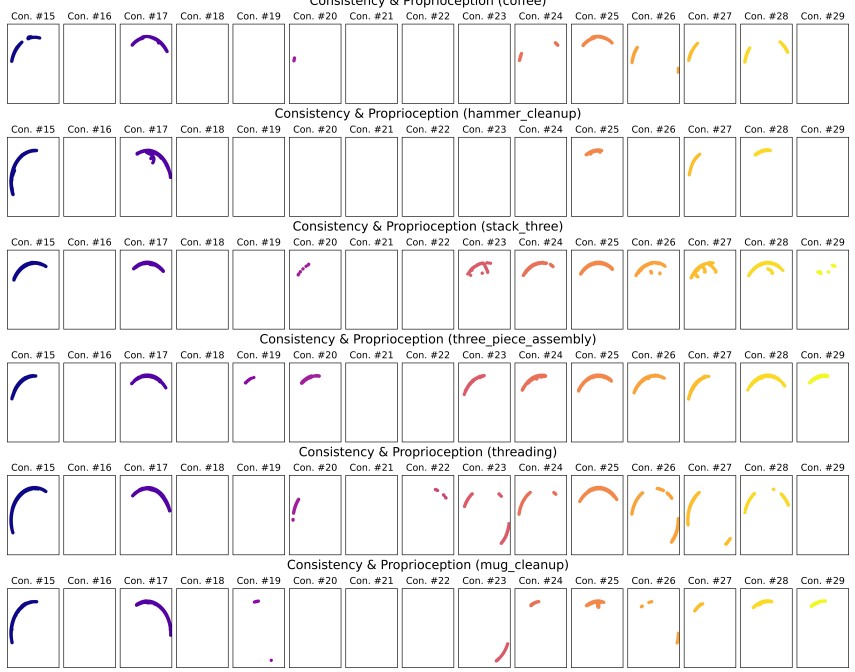

Figure 10: Consistency Visualization (part 2, manipulation concepts #15~29). Each row corresponds to a task and each column corresponds to a manipulation concept. The sub-figure at the intersection of a row and a column visualizes the PCA-reduced proprioceptive states from the task in the row that are associated with the manipulation concept in the column.

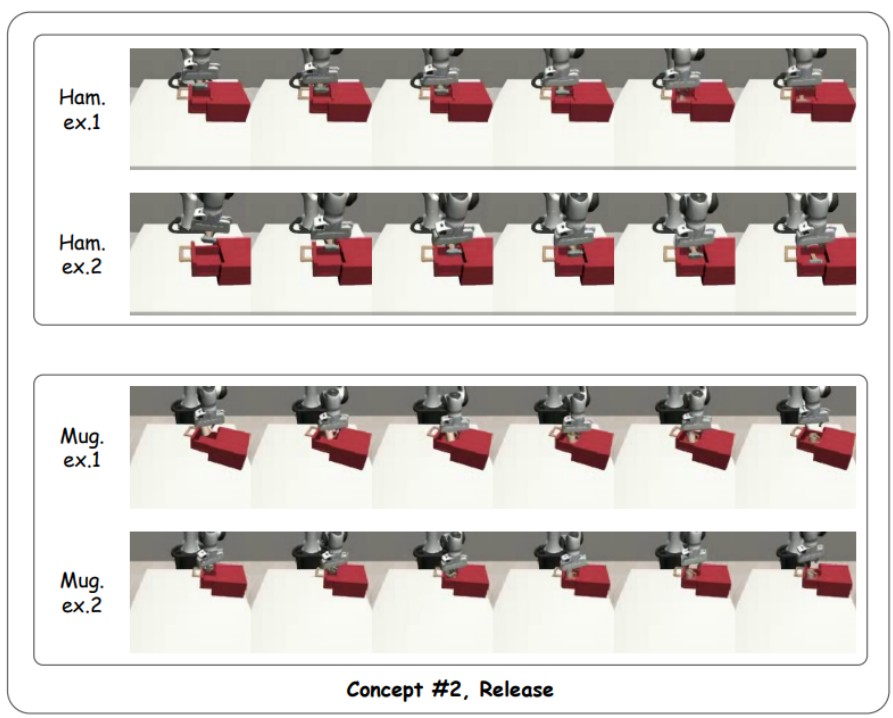

Figure 11: Consistency visualization of the concepts discovered by our method across different tasks. Here the robot arms release their grippers and put the objects down.

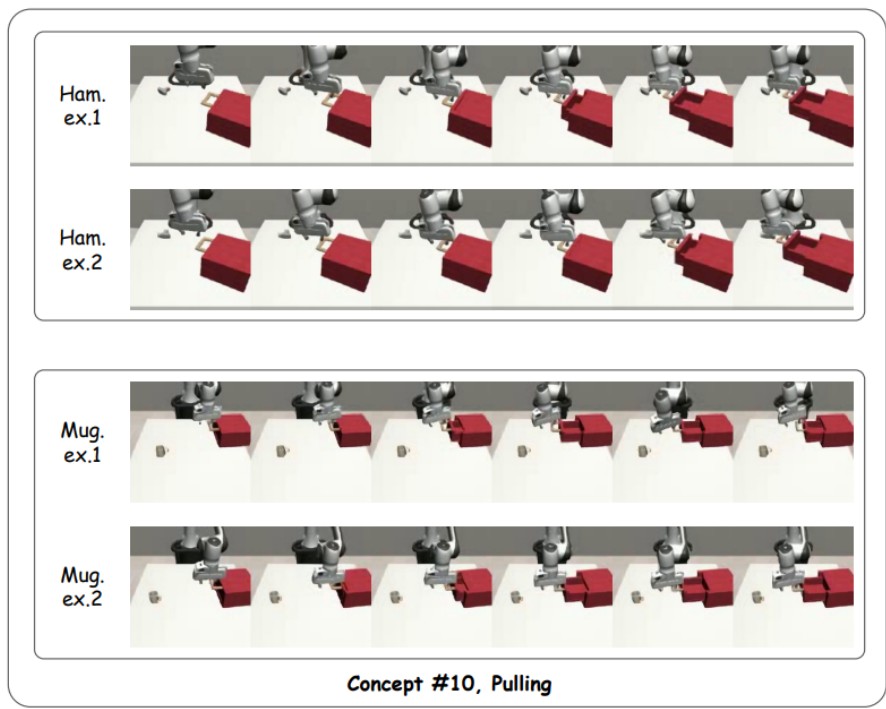

Figure 12: Consistency visualization of the concepts discovered by our method across different tasks. Here the robot arms pull things out.

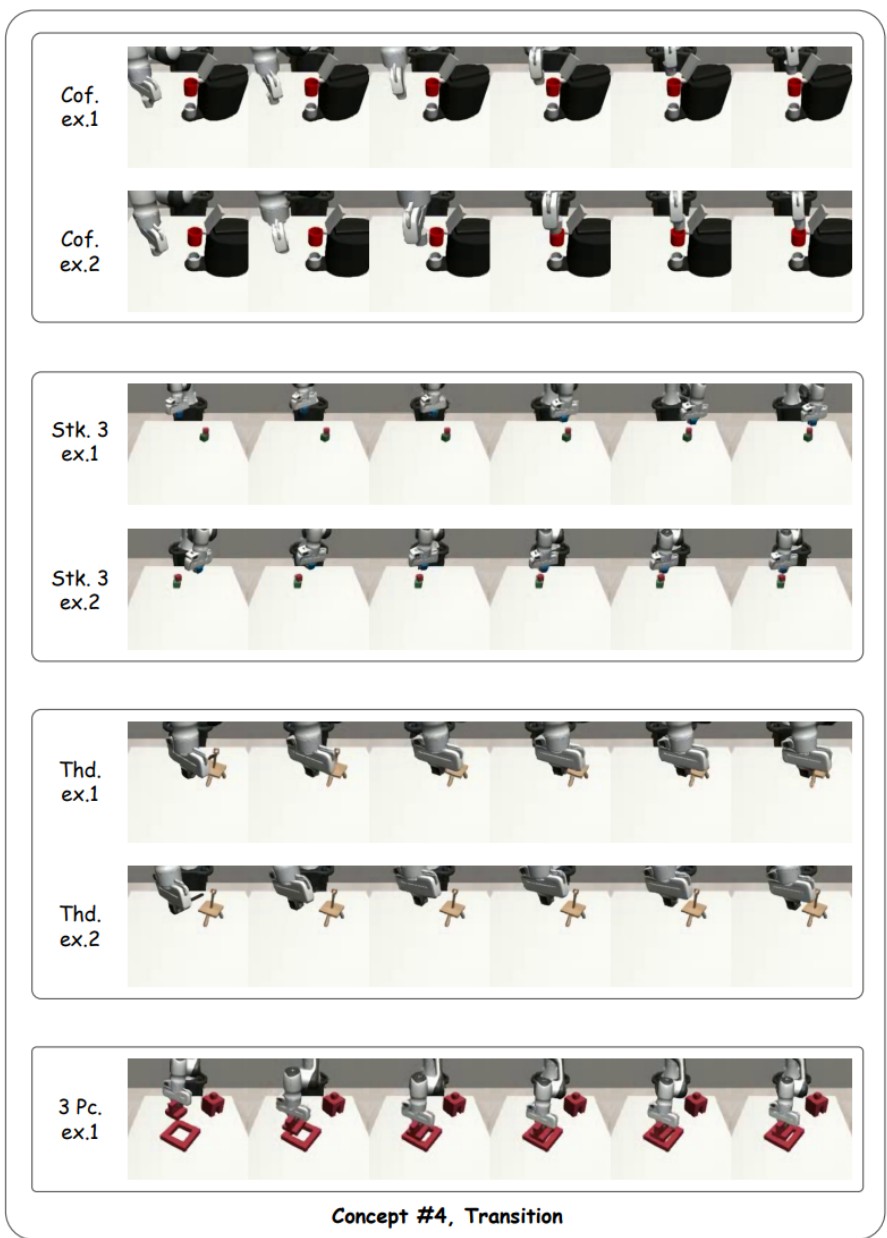

Figure 13: Consistency visualization of the concepts discovered by our method across different tasks. Here the robot arms transit the objects in hands.

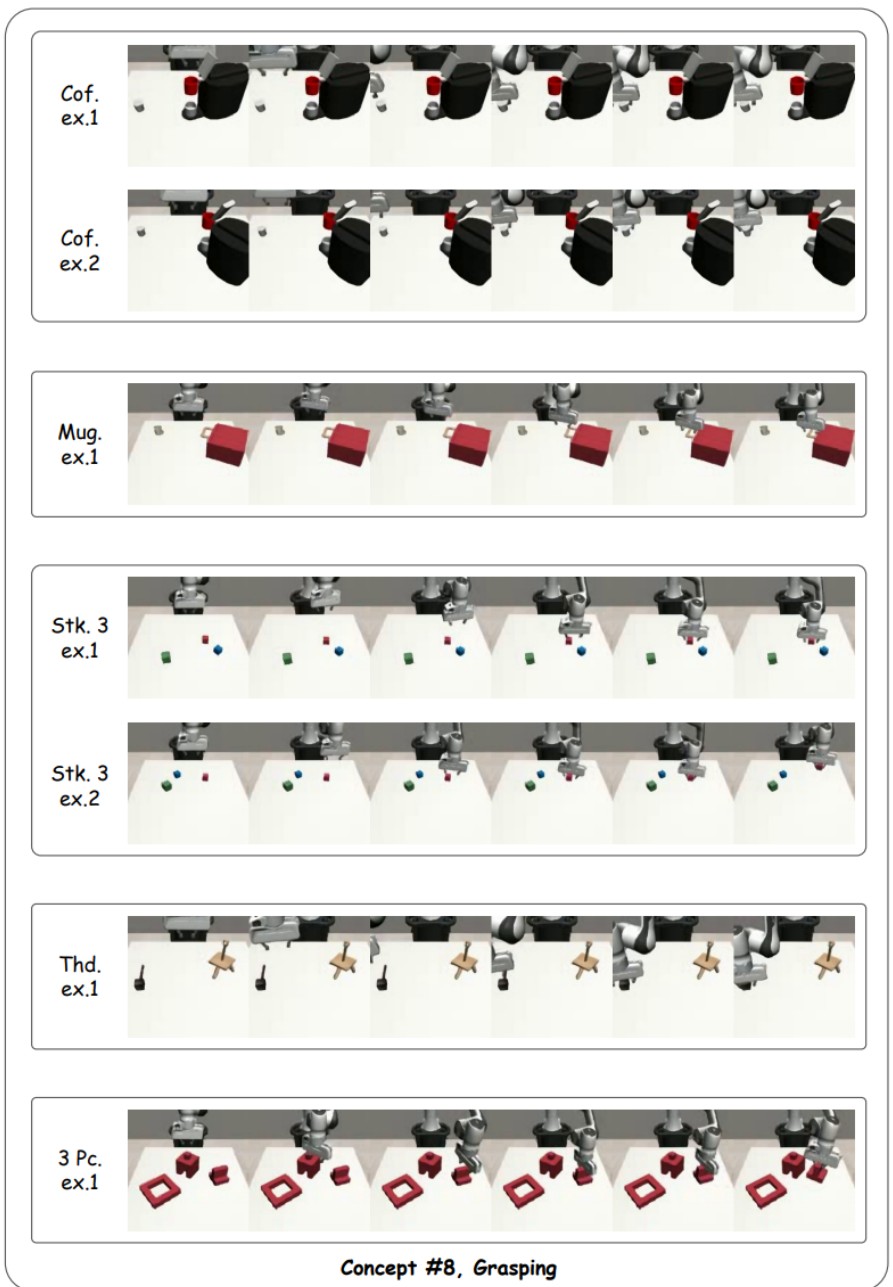

Figure 14: Consistency visualization of the concepts discovered by our method across different tasks. Here the robot arms move towards the objects and ready to pick.

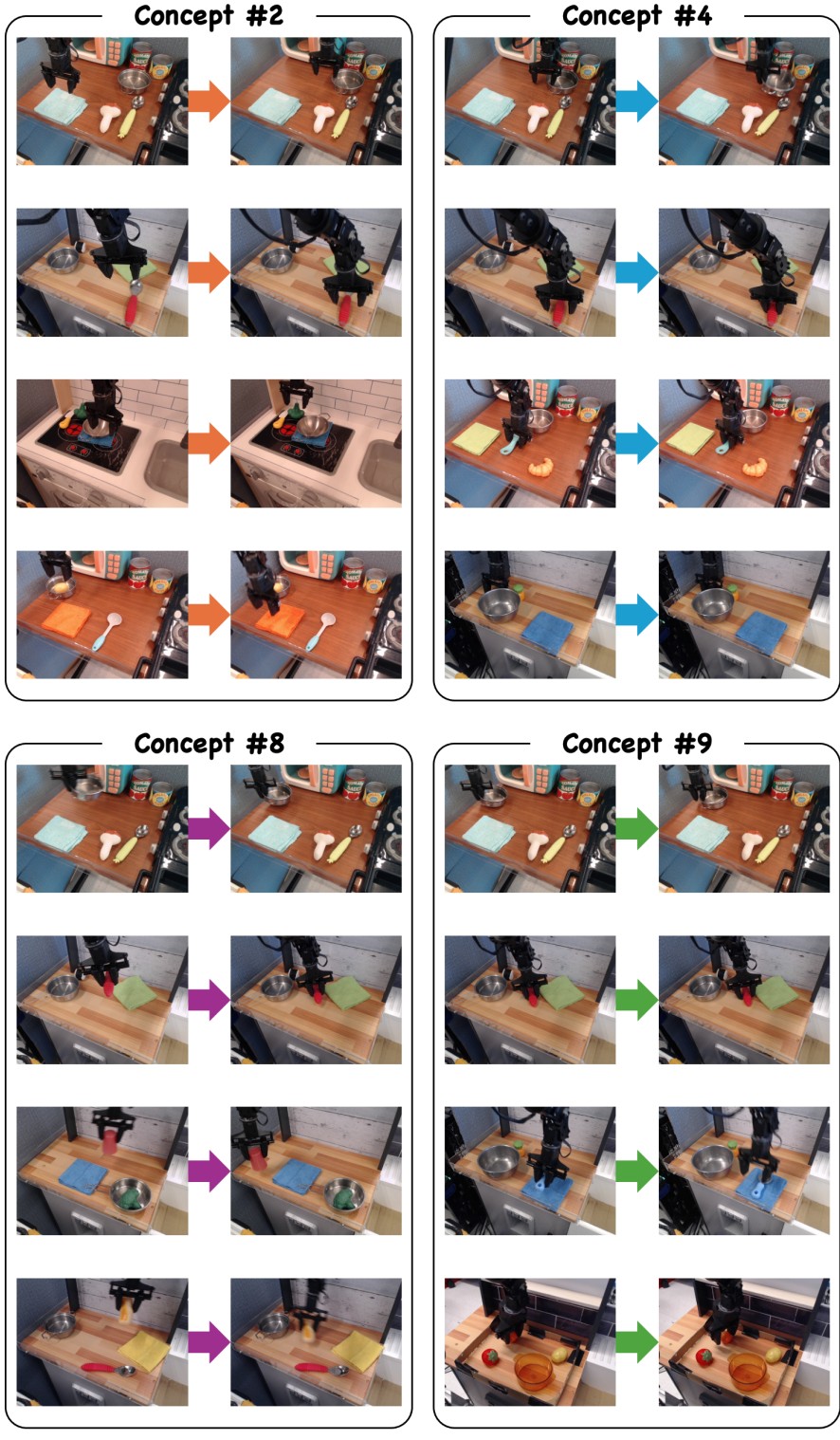

Figure 15: Concept Visualization of the Automatic Concept Discovery process (Sec. 3.1) applied to real-world data BridgeData V2 (Walke et al., 2023). A VQ-VAE with a codebook size of 10 is employed. Four discovered manipulation concepts are visualized: Concept #2 (reaching), Concept #4 (closing the gripper), Concept #8 (placing an object), and Concept #9 (opening the gripper to release an object).

