# OpenReview forum: "AutoCGP: Closed-Loop Concept-Guided Policies from Unlabeled Demonstrations"
_ICLR.cc/2025/Conference — ICLR 2025 Spotlight_

### Official Review · Reviewer_Umdm · 2024-10-17

**Soundness:** 4
**Presentation:** 4
**Contribution:** 3
**Rating:** 8
**Confidence:** 4

**Summary:**

The paper presents a novel imitation learning framework for training closed-loop concept-guided policies for robotic tasks, focusing on long-horizon tasks. The framework leverages self-supervised learning to autonomously discover manipulation concepts directly from robotic demonstrations, without the need for predefined skills or human-labeled data. The two main components of the system are:
- Automatic Concept Discovery, which identifies manipulation concepts by analyzing proprioceptive states and assigning discrete embeddings to different task segments.
- Concept-Guided Policy Learning, which utilizes these discovered concepts for policy learning. A Concept Selection Transformer (CST) selects relevant manipulation concepts in real time, while a Concept-Guided Policy (CGP) generates actions based on the selected concepts, using a diffusion policy to handle high-dimensional actions.

The framework is designed to dynamically adjust policies in response to environmental changes, ensuring robust performance across a variety of tasks. Experimental results show that the proposed method outperforms existing baselines in multiple robotic manipulation tasks, demonstrating improved generalization and performance in dynamic environments.

**Strengths:**

- The approach is highly original in combining concept discovery with closed-loop policy learning. Instead of relying on predefined, human-annotated labels or skills, the system autonomously discovers meaningful manipulation concepts directly from demonstrations. This removes the need for manual intervention and avoids misalignment between human semantics and robot operations.
The use of a self-supervised concept discovery mechanism is innovative, especially in its application to long-horizon robotic tasks. The introduction of a Concept Selection Transformer (CST) for dynamic task execution is novel and enables adaptive behavior based on feedback.
- The methodology is well-designed and robust, incorporating a clear pipeline for concept discovery and policy learning. The experimental setup is thorough, with comparisons to multiple baselines and a wide range of tasks, which demonstrates the effectiveness of the proposed framework.
The paper also includes an ablation study to assess the contributions of different components within the framework, which adds depth to the analysis and validation of the proposed method.
- The paper is clearly written, with detailed explanations of the method, supported by visualizations of the discovered concepts. The distinction between the two main components (Automatic Concept Discovery and Concept-Guided Policy Learning) is clearly articulated, and the figures help in understanding the flow of the framework.
The technical details of the system, including the loss functions and architecture choices, are well-explained.

**Weaknesses:**

- Concept Interpretability: While the system autonomously discovers manipulation concepts, it would be interesting to try to explore how interpretable or human-understandable these concepts are. It would also be interesting to see if all the discovered concepts are utilized in downstream policy learning and, if not, what the unused ones look like.

- Evaluation Scope: Although the paper evaluates the framework on a variety of tasks, the focus is primarily on tabletop manipulation in simulated environments. Expanding the evaluation to more complex morphologies or real-world scenarios would strengthen the claim that the approach generalizes well.

- Lack of Real-World Validation: While the framework performs well in simulation, it has not been validated on real-world robotic platforms. The paper acknowledges this limitation but does not provide a clear roadmap for transitioning the system to physical robots, which is crucial for demonstrating its practical applicability.

**Questions:**

- There is some line of work using mutual information maximization to discover concepts/skills from demonstration datasets [1, 2] or self-supervised methods [3]. It may help to include some insights into those works.
- While the annotations used in the method section are very stringent and scientifically sound, it can be confusing for readers with less background knowledge or patience to fully understand the message the paper tries to deliver.
- In the automatic concept assignment stage, is the encoder trained separately from other components, or are they trained simultaneously? If the former, how did you prevent the encoder from collapsing to map to the same codebook index?
- Are all the discovered concepts meaningful and physically possible? Can I condition on an arbitrary concept index and the generated trajectories are good to use, or is the Concept Selection Transformer still necessary to act as a filter to rule out discovered but useless concepts?
- How to deal with diversity? In skill discovery works, one common issue is a diverse set of skills/concepts can be discovered, but this diversity is canceled out by a downstream skill/concept selector due to its collapse on one or two most useful skills/concepts. How can you retain all the concepts discovered and demonstrate the diversity there?

[1] Li, C., Blaes, S., Kolev, P., Vlastelica, M., Frey, J. and Martius, G., 2023, May. Versatile skill control via self-supervised adversarial imitation of unlabeled mixed motions. In 2023 IEEE international conference on robotics and automation (ICRA) (pp. 2944-2950). IEEE.

[2] Peng, X.B., Guo, Y., Halper, L., Levine, S. and Fidler, S., 2022. Ase: Large-scale reusable adversarial skill embeddings for physically simulated characters. ACM Transactions On Graphics (TOG), 41(4), pp.1-17.

[3] Li, C., Stanger-Jones, E., Heim, S. and Kim, S., 2024. FLD: Fourier Latent Dynamics for Structured Motion Representation and Learning. arXiv preprint arXiv:2402.13820.

---

> ### Author Response · Authors · 2024-11-21
>
> Dear Reviewer Umdm,
>
> Thank you for your thoughtful and encouraging feedback on our work. We appreciate the recognition of our framework's originality in integrating concept discovery with closed-loop policy learning, particularly its ability to autonomously discover manipulation concepts without relying on human annotations. Also, thank you for your acknowledgment of the novelty and robustness of our Concept Selection Transformer (CST) and the thoroughness of our experimental validation, including the ablation studies. We are delighted that the clarity and depth of our methodology, technical details, and visualizations resonated with you. Next we address the questions and comments raised in the review.
>
> ***
> >**W.1** Concept Interpretability: While the system autonomously discovers manipulation concepts, it would be interesting to try to explore how interpretable or human-understandable these concepts are. It would also be interesting to see if all the discovered concepts are utilized in downstream policy learning and, if not, what the unused ones look like.
>
> **A to W.1**
>
> We sincerely appreciate your concern regarding the interpretability and utilization of the learned manipulation concepts.
>
> **1. Interpretability of Manipulation Concepts**
>
> To address the interpretability issue, we have presented visualizations of the manipulation concepts in Fig. 4, Fig. 5, and Fig. 11-14. Notably, the annotations in Fig. 5 and Fig. 11-14 assign semantic labels to each manipulation concept based on their correspondence to human interpretations in the annotations of the figures. These annotations highlight the resemblance between the manipulation concepts identified by our model and human-understandable semantics.
>
> **2. Utilization of Manipulation Concepts**
>
> Regarding the utilization of manipulation concepts, we would like to clarify that **all discovered manipulation concepts are utilized in downstream policy learning**, and we have provided a detailed explanation of how CST utilize these manipulation concepts in the latter **A to Q.4** and **A to Q.5**. In addition, we also want to clarify that **manipulation concepts’ applicability may vary depending on the specific task**. This variation aligns with human intuition; for instance, a concept such as "pulling" may not be relevant in certain subprocesses, such as "walking to the kitchen."
>
> To illustrate this, we analyzed the manipulation concepts used in two tasks: Coffee (coffee_d0, coffee_d1, coffee_d2) and Mug Cleanup (mug_cleanup_d0, mug_cleanup_d1). The results are summarized below:
>
> **Manipulation Concepts Used in Each Task**
> | Task | Manipulation Concepts |
> |:-:|:-:|
> | Coffee | 0, 1, 2, 3, 4, 5, 7, 8, 9, 10, 15, 17, 20, 24, 25, 26, 27, 28 |
> | Mug Cleanup | 0, 1, 2, 3, 4, 5, 7, 8, 9, 10, 15, 17, 19, 23, 24, 25, 26, 27, 28, 29 |
>
> **Manipulation Concepts Exclusively Used by Each Task**
> | Task | Manipulation Concepts |
> |:-:|:-:|
> | Coffee | 20 |
> | Mug Cleanup | 19, 23, 29 |
>
> We observed that the manipulation concepts unused in one task but employed in the other exhibit distinct proprioceptive state patterns and functionalities, explaining their task-specific relevance. Two examples are provided below:
> - **Manipulation Concept 20** (exclusive to the Coffee task): This concept corresponds to the action of closing the lid of a coffee maker. The robotic arm’s gripper performs a sub-circular motion to complete the operation. Example can be found at this path of the revised supplementary material: **Supplementary Material/rebuttal_visualizations/exclusive_concept/concept20_close_tap.mp4**
> - **Manipulation Concept 29** (exclusive to the Mug Cleanup task): This concept represents a straightforward "pushing" motion used to close a drawer, with the gripper following a linear trajectory. Example can be found at this path of the revised supplementary material: **Supplementary Material/rebuttal_visualizations/exclusive_concept/concept29_pushing_drawer.mp4**
> ***
> >**W.2** Evaluation Scope: Although the paper evaluates the framework on a variety of tasks, the focus is primarily on tabletop manipulation in simulated environments. Expanding the evaluation to more complex morphologies or real-world scenarios would strengthen the claim that the approach generalizes well.
>
> **A to W.2**
> Thank you for your inquiry regarding the scope of our evaluation. We acknowledge the limitations posed by our current equipment, which restrict us from performing real-world evaluations and evaluation on more complex morphologies. However, we are actively exploring the effectiveness of our methods (like human pose estimation) within our current capabilities. We appreciate your understanding and are committed to expanding our evaluation scope as resources allow.

---

> ### Author Response · Authors · 2024-11-21
>
> >**W.3** Lack of Real-World Validation: While the framework performs well in simulation, it has not been validated on real-world robotic platforms. The paper acknowledges this limitation but does not provide a clear roadmap for transitioning the system to physical robots, which is crucial for demonstrating its practical applicability.
>
> **A to W.3**
> Thank you for your interest in the roadmap for real-world implementation of our methodologies. We believe that the Automatic Concept Discovery process (Sec. 3.1) holds the potential of identifying manipulation concepts from a diverse array of real-world robotic data. Furthermore, the policy formation based on these manipulation concepts can be effectively executed in a manner similar to the imitation learning approach in Sec. 3.2 for real work robotic data, utilizing the discovered concepts.
>
> To showcase the potential of our method, we applied the Automatic Concept Discovery process to BridgeDataV2(https://rail-berkeley.github.io/bridgedata/) and visualized some discovered manipulation concepts in **Fig. 15** of the revised manuscript. Our findings reveal that the consistency of proprioceptive states across different manipulation scenarios is preserved, underscoring the robustness of our concept discovery methodology.
>
> Looking ahead, we anticipate that leveraging a larger and more diverse robotics dataset will further enhance the discovery of varied and effective manipulation concepts. This advancement would, in turn, strengthen the guidance provided by these concepts for the closed-loop, concept-guided policy discussed in Sec. 3.2.
> ***
> >**Q.1** There is some line of work using mutual information maximization to discover concepts/skills from demonstration datasets [1, 2] or self-supervised methods [3]. It may help to include some insights into those works.
>
> **A to Q.1**
> We sincerely appreciate the reviewer pointing out these valuable related works on mutual information maximization for concept/skill discovery. We have incorporated these references and their key insights into our updated Sec. 2 Related Work, which helps provide a more complete context for our work.
> ***
> >**Q.2** While the annotations used in the method section are very stringent and scientifically sound, it can be confusing for readers with less background knowledge or patience to fully understand the message the paper tries to deliver.
>
> **A to Q.2**
> Thanks for this helpful feedback. We will revise **Sec.3** to make our method description more accessible and reader-friendly in our final version.
> ***
> >**Q.3** In the automatic concept assignment stage, is the encoder trained separately from other components, or are they trained simultaneously? If the former, how did you prevent the encoder from collapsing to map to the same codebook index?
>
> **A to Q.3**
> Thank you for your question regarding the training of the encoder in Eq.1.
>
> For a detailed understanding of the training dynamics within a single iteration of the Automatic Concept Discovery process (Sec. 3.1), please refer to the pseudocode provided in Sec. A.4. The formation of concepts that fulfill the features described in Sec. 3.1 (including goal state detection, goal state evaluation, and goal consolidation) requires that the encoder loss ($L^{\text{vq}}$) be trained in conjunction with the predictive decoder losses ($L_{\text{gd}}$ $L_{\text{ge}}^a$ $L_{\text{ge}}^c$ $L_{\text{gc}}$) (Also see Eq. 9).
>
> Regarding the common issue of collapse in VQ-VAE models, we have observed that the inclusion of $L_{\text{gc}}$ (goal consolidation loss) assists in preventing this phenomenon. To illustrate this, we have provided statistics comparing the number of discovered manipulation concepts with and without the use of $L_{\text{gc}}$ below (we run the experiments 10 times and provide the average number here).
>
> | | Ours | w/o GC |
> | :-: | :-: | :-: |
> | Num. of Manipulation Concepts | 23.7 | 6.3 |
>
> These figures clearly demonstrate the effectiveness of $L_{\text{gc}}$ in maintaining the diversity of the concepts discovered.
> ***
> >**Q.4** Are all the discovered concepts meaningful and physically possible? Can I condition on an arbitrary concept index and the generated trajectories are good to use, or is the Concept Selection Transformer still necessary to act as a filter to rule out discovered but useless concepts?
>
> **A to Q.4**
> Thank you for your question related to the manipulation concepts’ meaningfulness and validity, as well as the role of Concept Selection Transformer (CST in Sec. 3.2).
>
> Note that all our concepts are discovered in the multitask dataset, and each concept represents a repeated motion pattern, in certain tasks. Thus, the manipulation concepts are all meaningful, but each concept will only be understandable and used in its corresponding tasks and motions.

---

> ### Author Response · Authors · 2024-11-21
>
> In regards to the usage of Concept Selection Transformer (CST), we want to clarify that CST is the module that makes the concept-guided policy (Sec. 3.2) align with the manipulation concepts discovered from the Automatic Concept Discovery (Sec. 3.1) process. For every time-step in a manipulation process, this alignment provides the policy with a manipulation concept, featuring the motor skill needed to carry out. Thus, CST does not ‘filter out useless concepts’, but rather it maps current observation with corresponding concepts, given the fact that all the concepts are useful in their own regions.
>
> ***
> >**Q.5** How to deal with diversity? In skill discovery works, one common issue is a diverse set of skills/concepts can be discovered, but this diversity is canceled out by a downstream skill/concept selector due to its collapse on one or two most useful skills/concepts. How can you retain all the concepts discovered and demonstrate the diversity there?
>
> **A to Q.5**
> Thank you for your question related to the diversity of manipulation concepts.
>
> Related to our response to you in **A to Q.4**, the diversity of manipulation concepts we discovered is only related with the Automatic Concept Discovery (Sec. 3.1) process itself. In our experiment, the diversity is not canceled out, because the concept selector is a well-trained transformer and doesn’t collapse on one or two most concepts.
>
> We employ Shannon Entropy to quantify the diversity in CST's output distribution. A high entropy value indicates that the concept selector utilizes a broad range of concepts rather than collapsing to a few dominant ones. Specifically, if the concept selector were to concentrate solely on one or two concepts, the Shannon Entropy would fall within [0, 1]. Our experimental results demonstrate that the Shannon Entropy consistently exceeds 1.5, indicating that our concept selector effectively leverages a diverse set of concepts for downstream policy learning.
>
> | Tasks | Coffee d2 | Mug cleanup d1 |
> |:-:|:-:|:-:|
> | Shannon Entropy | 1.6241 | 2.4504 |
> ***
> To recap, we have made effort to address the weaknesses and questions raised in the review. We conducted a detailed study on the application of our manipulation concepts across different tasks.We outlined the potential and provided a roadmap for implementing our methodology in more complex and real-world scenarios. We reviewed the relevant works provided, incorporating and citing them where appropriate. Lastly, we clarified the training details of the Automatic Concept Discovery process (Sec. 3.1) and elaborated on the role of the Concept Selection Transformer (CST), specifically CST’s use of manipulation concepts discovered during the Automatic Concept Discovery process.
>
> We hope our response can address your questions. If you have any further questions, please don't hesitate to contact us.
>
> Thanks,
>
> The Authors
> ***
> [1] Li, C., Blaes, S., Kolev, P., Vlastelica, M., Frey, J. and Martius, G., 2023, May. Versatile skill control via self-supervised adversarial imitation of unlabeled mixed motions. In 2023 IEEE international conference on robotics and automation (ICRA) (pp. 2944-2950). IEEE.
> [2] Peng, X.B., Guo, Y., Halper, L., Levine, S. and Fidler, S., 2022. Ase: Large-scale reusable adversarial skill embeddings for physically simulated characters. ACM Transactions On Graphics (TOG), 41(4), pp.1-17.
> [3] Li, C., Stanger-Jones, E., Heim, S. and Kim, S., 2024. FLD: Fourier Latent Dynamics for Structured Motion Representation and Learning. arXiv preprint arXiv:2402.13820.

---

> > ### Comment · Reviewer_Umdm · 2024-11-27
> > **Thanks**
> >
> > Thank you for addressing the concerns and questions raised. With the answers given by authors properly integrated in the final paper, the its strengths and position can be further enhanced.

---

> > > ### Author Response · Authors · 2024-11-28
> > > **Thanks again for your valuable feedback**
> > >
> > > Dear Reviewer Umdm,
> > >
> > > We are glad that the raised concerns and questions have been addressed in the rebuttal. We will incorporate these answers in the final paper. And thanks again for helping us improve the quality of our work.
> > >
> > > Thanks,
> > >
> > > The Authors

---

### Official Review · Reviewer_Wk2p · 2024-10-26

**Soundness:** 4
**Presentation:** 4
**Contribution:** 3
**Rating:** 8
**Confidence:** 3

**Summary:**

This paper presents a method for discovering robot actions and manipulation policies from unlabeled demonstration data. It presents techniques for discovering generic action/manipulation concepts and their goals, evaluating progress towards those goals, and generating policies to achieve the goals. This is done using various modern machine learning techniques and problem/loss function formulations. For concept discovery, A VQ-VAE is used to discover concepts and put them into a finite codebook. A transformer model is used for goal state detection. A hypernetwork and another transformer are used for goal state evaluation, and a final transformer is used for goal consolidation. All of these networks are jointly trained using a large combined loss function. Then, concept selection and policy generation are learned using a combination of a transformer and a diffusion policy. Finally, the approach is validated by training on a set of 950 robot demonstrations in 6 different tasks, then evaluated with randomized starting positions (all in simulation). The success rate of different tasks learned via the proposed method is higher than for competing metrics, even/especially in the presence of perturbing noise.

**Strengths:**

* The paper compares the proposed approach to a set of highly relevant prior work in a fairly direct head-to-head comparison and the proposed approach nearly universally outperforms the prior work by a good margin.

* The approach is a novel combination of several modern techniques to solve a complex open problem in robotics. It expands upon work such as InfoCon by adding policy learning directly into the framework of manipulation concepts.

* The paper is written clearly and the construction of the experiments is well-explained (training, structure of network, etc.), including the re-implementation of baselines.

**Weaknesses:**

* The primary weakness of this study is that it only contains a brief evaluation of the quality of the learned skills (i.e., are they generalized, are they more generalized than prior approaches), and a single visual example. Characterizing XSkill "concepts" via K-means clustering is perhaps not the most charitable interpretation of their concepts, as K-means clustering will naturally tend to collect noise in its clusters (noise is one of the stated drawbacks of the xskill results). Ideally there would be some sort of mean similarity score or some other comparison method - even something like a 2D embedding visualization of the different skills and concepts, if possible to generate, would give better faith in the stated quality of the learned concepts. This is a conference about learning representations, so I would love to see more focus on the quality of the representation of "concept". That being said, other papers, including InfoCon, also use final task success as the main comparison metric, so I don't consider this a big enough weakness to reject the paper.


* It is not clear why such a large variety of approaches were used in this paper. Many different transformer sizes were used, in addition to the diffusion policy and the VQ-VAE. Specifically, the use of hypernetworks also seems like an unnecessary addition to what is already a diverse and expressive set of architecture choices.

**Style notes**

* perhaps this is a notational convention I am unaware of, but the parentheses in epsilon^(n) seem like unnecessary clutter.

* Should eq 11 be negative to account for the fact that the log term will always be negative?

* In eq 10, is the selected action from pi_D conditioned on the k selected by p_CST? If so, this could be made clearer, perhaps by splitting into two equations.

**Questions:**

* At some point, the number of different networks + their parameters runs a risk of over-parameterization and overfitting. This is especially true since it seems that the set of demonstrations were collected over the same set of tasks as used in policy generation and success measurement. Combined with the lack of qualitative cross-task concept comparison or similarity analysis, I do have overfitting worries here. I have no evidence to directly support this flaw other than my own experience training large robotics models, but would still like to know the authors' thoughts on how overfitting is avoided in the proposed approach.

* Why was a hypernetwork used for a single piece of the approach?

* How were the 950 demonstrations in the experiment generated, and were they over all 6 tasks, or a different set/subset of tasks? I was not able to find this anywhere in the paper.

---

> ### Author Response · Authors · 2024-11-21
>
> Dear Reviewer Wk2p,
>
> Thank you for recognizing the strengths of our paper. We appreciate your acknowledgment of our novel approach in using advanced machine learning techniques for discovering robot actions and manipulation policies from unlabeled data. Your acknowledgment of our method's systematic integration of VQ-VAE, transformers, and hypernetwork is encouraging. We also appreciate your positive feedback on our method's robustness and effectiveness, demonstrated by higher success rates in diverse task simulations. Next we address your comments and questions in sequence.
> ***
> >**W.1** Unclear presentation
> The primary weakness of this study is that it only contains a brief evaluation of the quality of the learned skills (i.e., are they generalized, are they more generalized than prior approaches), and a single visual example. Characterizing XSkill "concepts" via K-means clustering is perhaps not the most charitable interpretation of their concepts, as K-means clustering will naturally tend to collect noise in its clusters (noise is one of the stated drawbacks of the xskill results). Ideally there would be some sort of mean similarity score or some other comparison method - even something like a 2D embedding visualization of the different skills and concepts, if possible to generate, would give better faith in the stated quality of the learned concepts. This is a conference about learning representations, so I would love to see more focus on the quality of the representation of "concept". That being said, other papers, including InfoCon, also use final task success as the main comparison metric, so I don't consider this a big enough weakness to reject the paper.
>
> **A to W.1**
> Thank you for your thoughtful and insightful comments regarding the direct evaluation of the quality of representation for manipulation concepts.
>
> In response to your concern, we have conducted a new visualization study inspired by the approach in [https://arxiv.org/abs/2410.11758](https://arxiv.org/abs/2410.11758). This study, presented in **Fig. 9 and 10** in **Sec. C.3** of the revised manuscript, examines the consistency of our manipulation concepts in relation to the proprioceptive states of the robots. The results indicate that our methodology effectively captures consistent manipulation concepts across various tasks.
>
> We provide further detail on the visualization. In Fig. 9 and 10, each row corresponds to a task, and each column represents a specific manipulation concept. The sub-figure at the intersection of a given row and column visualizes all the proprioceptive states from the task in the row that are associated with the manipulation concept in the column. To facilitate visualization, we applied PCA to the proprioceptive states across all tasks, and the visualized points represent the PCA-reduced representations. The results show that non-empty sub-figures within each column exhibit similar patterns, emphasizing the consistency of the learned manipulation concepts across different tasks.
>
> That said, we acknowledge a limitation of this visualization method. For manipulation concepts associated with broader ranges of proprioceptive state patterns, the visualizations may become less informative. In such cases, the points tend to densely populate the entire space, leading to distributions that are challenging to interpret. Another limitation arises when the manipulation concepts are represented as feature vectors rather than discrete symbols. While the method is well-suited for naturally discrete symbols, feature vectors—where each component can take on continuous values—require clustering to convert them into symbolic representations (i.e., discrete symbols representing each cluster). This clustering step introduces potential subjectivity and bias, depending on the choice of clustering method.
>
> Looking ahead, we aim to explore more nuanced approaches to evaluate and analyze manipulation concepts, developing assessments that offer deeper and more persuasive insights.
> ***
> >**W.2** It is not clear why such a large variety of approaches were used in this paper. Many different transformer sizes were used, in addition to the diffusion policy and the VQ-VAE. Specifically, the use of hypernetworks also seems like an unnecessary addition to what is already a diverse and expressive set of architecture choices.
>
> **A to W.2**
> Thank you for your inquiry regarding our architectural choices.
>
> Most of the structural designs employed in our architecture are derived from established works. For example, the incorporation of designs such as Hypernetworks, VQ-VAE, and the number of Transformer layers, is adapted from the InfoCon (https://arxiv.org/abs/2404.10606) design. We have made slight adjustments to these components to enhance the performance for our experiments. Also, the implementation of the Diffusion Policy was influenced by its application in the MimicGen environment of our paper.

---

> ### Author Response · Authors · 2024-11-21
>
> In addition to leveraging proven techniques, we carefully considered the rationale behind continuing with certain designs. For example, the use of a Hypernetwork was primarily driven by the need to efficiently form the value function $\mathcal{V}$ as specified in Eq.5. Using a separate neural network for each of the $K$ manipulation concepts would be parameter-intensive. Therefore, we opted for a more parameter-efficient approach that conditions on the manipulation concepts. This decision also maintains a balance between data adaptability and parameter efficiency, as discussed in the referenced paper on the benefits of HyperNetworks (available at [https://arxiv.org/abs/2306.06955](https://arxiv.org/abs/2306.06955)).
>
> Furthermore, the choice of the Diffusion Policy was informed by its robust capability to handle highly variable demonstrations provided by the MimicGen Demos, such as those tasks with larger subscript of $D$ (e.g. Coffee $D_2$). The strong distribution capture capability of diffusion models makes them an ideal choice for such scenarios.
>
> Regarding our use of VQ-VAE, it naturally provides discrete symbolic labels based on index, which facilitates the process of identifying the completion of manipulation concept processes. By labeling each timestep with a manipulation concept from the VQ-VAE codebook, we can naturally delineate continuous timesteps under the same label as part of a subprocess and identify the transition between subprocesses when the VQ-VAE label changes. For a detailed explanation, please refer to Goal State Detection in Sec. 3.1.
> ***
> >W.3 Style notes
> >- perhaps this is a notational convention I am unaware of, but the parentheses in epsilon^(n) seem like unnecessary clutter.
> >- Should eq 11 be negative to account for the fact that the log term will always be negative?
> >- In eq 10, is the selected action from pi_D conditioned on the k selected by p_CST? If so, this could be made clearer, perhaps by splitting into two equations.
>
> **A to W.3:**
> Thank you for bringing these style-related issues to our attention. Regarding the notation $\epsilon^{(n)}$, we have retained this format since it explicitly denotes different noise levels at different timesteps n, which we believe is important for clarity. We have carefully addressed the remaining points and revised the manuscript accordingly to ensure greater clarity.
> ***
> >**Q.1** At some point, the number of different networks + their parameters runs a risk of over-parameterization and overfitting. This is especially true since it seems that the set of demonstrations were collected over the same set of tasks as used in policy generation and success measurement. Combined with the lack of qualitative cross-task concept comparison or similarity analysis, I do have overfitting worries here. I have no evidence to directly support this flaw other than my own experience training large robotics models, but would still like to know the authors' thoughts on how overfitting is avoided in the proposed approach.
>
> **A to Q.1**
> Thank you for raising the important issue of handling overfitting in our study.
>
> We would like to address your concern by discussing how overfitting is avoided in our proposed approach through two key aspects: Automatic Concept Discovery and Policy Learning.
>
> **1. Why Automatic Concept Discovery Mitigates Overfitting**
>
> We outline below the reasons and evidence demonstrating that the Automatic Concept Discovery process (Sec. 3.1) effectively mitigates overfitting.
>
> **1.1. Blending of Tasks in Automatic Concept Discovery**
>
> Our Automatic Concept Discovery process leverages demonstrations from a variety of tasks without incorporating task-specific information such as task names or descriptions (as detailed in Implementation Details, Sec. 4.1). This ensures that the discovered manipulation concepts are not overly influenced by the specifics of any single task and can be applied across a variety of settings. By blending data from multiple tasks and excluding task-specific information, we aim to prevent the model from overfitting to narrow, task-specific manipulation concepts.
>
> **1.2. Focus on Proprioceptive States in Concept Discovery.**
>
> As discussed in our response to **Reviewer LkiS’s Q.1** (A to Q.1) and elaborated in Sec. A.1, we focus exclusively on proprioceptive states during concept discovery. This design choice allows the model to concentrate on features derived from the robot's motion patterns, effectively mitigating overfitting to environmental nuances or background contexts that might arise from full environmental images. Our visualizations (Fig. 5 and Sec. C.3) demonstrate that the learned manipulation concepts consistently capture key motion patterns across tasks,  rather than fitting on some unrelated environmental feature differences.

---

> ### Author Response · Authors · 2024-11-21
>
> **1.3. Abstraction of Sub-processes**
>
> The Average Segment Length metric (Tab. 3, Sec. C.2) demonstrates that our manipulation concepts serve as meaningful abstractions of underlying processes rather than overfitted representations of the proprioceptive data. This metric measures the average duration (in continuous time steps) of segments labeled with the same manipulation concept. To further illustrate the abstraction achieved through our Automatic Concept Discovery process, we present a portion of Tab. 3 below, showing the Average Segment Length of our manipulation concepts:
>
> |Tasks| Cof. $D_0$ | Cof. $D_1$ | Cof. $D_2$ | Ham. $D_0$ | Ham. $D_1$ | Stk. $D_0$ | Stk. $D_1$ | 3 Pc. $D_0$ | 3 Pc. $D_1$ | Thd. $D_0$ | Mug $D_0$ | Mug $D_1$ |
> | :-: | :-: | :-: | :-: | :-: | :-: | :-: | :-: | :-: | :-: | :-: | :-: | :-: |
> | Average Segment Length | 28.3 | 31.0 | 27.6 | 33.9 | 29.5 | 26.7 | 22.5 | 28.5 | 28.8 | 33.3 | 29.3 | 28.5 |
>
> In summary, the strategies incorporated in the Automatic Concept Discovery process, combined with the presented evidence, demonstrate its robustness against overfitting.
>
> **2. How Policy Learning Alleviates Overfitting**
>
> Policy learning in our approach is exposed to the extensive variability in demonstrations provided by the MimicGen environment. This variability serves as a critical mechanism for mitigating overfitting during imitation learning, as elaborated in our response to **Reviewer UouH’s W.1**, second concern. By exposing the model to a diverse range of scenarios, including challenging and non-ideal conditions, the training process ensures that the policies are not trained on a limited set of conditions, thereby mitigating the risk of overfitting during policy learning.
> ***
> >**Q.2** Why was a hypernetwork used for a single piece of the approach?
>
> **A to Q.2**
> Thank you for your inquiry about the Hypernetwork. For a detailed explanation of our rationale for using it, please refer to our response to you under **A to W.2**.
> ***
> >**Q.3** How were the 950 demonstrations in the experiment generated, and were they over all 6 tasks, or a different set/subset of tasks? I was not able to find this anywhere in the paper.
>
> **A to Q.3**
> Thank you for your question regarding the demonstrations used in our experiments. We have employed 950 training demonstrations from the MimicGen benchmark for each task we selected and their respective levels of initial setting variation. To provide further clarity on this matter, please refer to the Implementation details section in Sec. 4.1 of our revised manuscript.
> ***
> To recap, we have made effort to address the weaknesses and questions raised in the review. We provided a new visualization experiment to clarify the consistency of our manipulation concepts. We also explained the rationale behind our architectural choices, offering insights into the reasoning behind these decisions. We have revised the manuscript to enhance the clarity of the notation, refine the descriptions of the methodology, and detail the experimental settings more precisely. Finally, we provided insights into how the proposed methods mitigate overfitting in both the Automatic Concept Discovery process and policy learning.
>
> We hope our response can address your questions. If you have any further questions, please don't hesitate to contact us.
>
> Thanks,
>
> The Authors

---

> ### Comment · Reviewer_Wk2p · 2024-11-22
> **thank you for the detailed responses**
>
> Thank you very much for your responses. I don't feel comfortable raising my rating even higher as I still think that the overall approach is unnecessarily complex, particularly in the number of different learning techniques used. But the responses and edits definitely help strengthen the paper's position and address my comments.

---

> > ### Author Response · Authors · 2024-11-23
> >
> > Thank you for your thoughtful feedback. We understand your concern about complexity. While our approach does involve several learning techniques, each component serves a distinct and necessary functional purpose in our system, as our anstwer to weakness 2. We appreciate your recognition of how our revisions have strengthened the paper. We'll continue exploring ways to balance functionality and simplicity in our future work.

---

### Official Review · Reviewer_UouH · 2024-11-04

**Soundness:** 3
**Presentation:** 2
**Contribution:** 2
**Rating:** 8
**Confidence:** 3

**Summary:**

## Research Question
For imitation learning(IL), 1)data annotation requires additional human effort; 2) the inherent subjectivity in manual data collection\annotation leads to manipulation concepts that do not align well with the robot’s configuration, and such inconsistency will accumulate, and 3)  IL will face further difficulties due to the varying and unpredictable conditions encountered at each step.


## Proposed Method
To mitigate the above challenges, the author proposed a framework that combines automatic concept discovery with closed-loop concept-guided policy learning. Such a framework will take unlabeled demonstration for training and autonomously extract and utilize manipulation concepts directly from the robot’s proprioceptive states.  Specifically, for training, the "Automatic Concept Discovery" module will first derive manipulation concepts automatically. Then, for both training and testing, the "Concept Selection Transformer (CST)" will propose and adjust manipulation concepts in real time during the robot’s interactions with the environment. Finally, the "Concept-Guided Policy (CGP)" utilizes the selected manipulation concepts to execute actions based on instantaneous visual input

**Strengths:**

1. The idea is a good catch for research questions

The manual annotation(sample efficiency), accumulative action inconsistency, and environmental unpredictability are classic bottlenecks for imitation learning,  and the general idea of "self-discovered sub-policy\concepts" is also not something new. However, the author directly extracts the concepts from the robot’s proprioceptive states,  such design not only mitigates the misalignment caused by ambiguities in human semantics but also adapts dynamically to unforeseen situations.

2. Appropriate baseline and good performance

The selected baselines are new and SOTA, and reasonable to compare with the proposed method.  Major advances of the proposed method are observed according to Table 1.

**Weaknesses:**

1. Unclear presentation

The figures are a little bit messy (e.g. lack the connection of each part in Fig 3, the Closed-Loop Policy part in Fig 2 is unclear  ), but it's a minor point. What matters are: 1) if the discovered concepts are consistent in all demos, i.e. if the concepts are in a unified, task-agnostic space for all demos; 2) For evaluation, it says 950 demos for each task, so does it mean: 2.1) the training data is 950 * |# of task| for single unified policy (set of concepts) and get tested all 6 type of tasks at once( in a mixed way). Or: 2.2) for each task,  it uses 950 demos to train and test only for the task.

For 1), I suppose the concepts are consistent in all demos according to the formula (1), but it's better to be clear at the beginning.  For 2), I'm not sure if it's 2.1) or not, if it's 2.2), the performance will be questionable as it's very close to overfitting.

2. Less practical problem setting

There was a trend to label the raw robot demonstration automatically, but a more practical problem is the raw demo collection is much more expensive than demo annotation, and therefore, some researchers turn to human-play data to get more data to use. And it's the same story for this paper, the expensive robot demo collection reduces the significance of this work

3. Concern about generalizability and scalability

Though the paper claims better performance (compared with baselines) in a more diversified initial environment,  here the diversity is only expressed by object positions and placement angles. Some concerns are: 1)What if there is concept inconsistency in the demos,  e.g. to do the same manipulation, in a different way; 2)With the growth of the demo amount, will the # number of concepts converge or grow fast;

**Questions:**

Dear author:

Generally, the paper is a decent work, but it needs several extra explanations and explorations.

And I will raise my rating if the following concerns are solved:
1. The concerns in the "Weakness" part.
2. A minor point is to add an IL SOTA baseline specific to the problem, which is not limited to a Diffusion Policy basis.

---

> ### Author Response · Authors · 2024-11-21
>
> Dear Reviewer UouH,
>
> Thank you for recognizing the strengths of our proposed framework in addressing the challenges of imitation learning (IL), e.g., human effort needed for annotation and inherent subjectivity in manual data annotation. We appreciate your acknowledgment of how our method autonomously extracts and effectively utilizes manipulation concepts from the robot’s proprioceptive states, reducing manual annotation efforts and adapting dynamically to environmental interactions. Furthermore, thanks for confirming the solidity of our experiments and the significance in the improvements. In the following, we address your concerns and questions.
> ***
> >**W.1** Unclear presentation. The figures are a little bit messy (e.g. lack the connection of each part in Fig 3, the Closed-Loop Policy part in Fig 2 is unclear ), but it's a minor point. What matters are: 1) if the discovered concepts are consistent in all demos, i.e. if the concepts are in a unified, task-agnostic space for all demos; 2) For evaluation, it says 950 demos for each task, so does it mean: 2.1) the training data is 950 * |# of task| for single unified policy (set of concepts) and get tested all 6 type of tasks at once( in a mixed way). Or: 2.2) for each task, it uses 950 demos to train and test only for the task. For 1), I suppose the concepts are consistent in all demos according to the formula (1), but it's better to be clear at the beginning. For 2), I'm not sure if it's 2.1) or not, if it's 2.2), the performance will be questionable as it's very close to overfitting.
>
> **A to W.1**
> Thank you for your insightful questions regarding the features of the manipulation concepts discovered and the experimental setup used in our study. We greatly appreciate your interest and are eager to address concerns 1) and 2) in a sequential manner.
>
> **Regarding concern 1)**, our Automatic Concept Discovery process (Sec. 3.1) is implemented on demonstrations across various tasks without the use of specific task-related information such as task names and descriptions (Please refer to the Implementation details in Sec. 4.1 for more specifics). This strategy enables the discovery of manipulation concepts that are applicable across sub-processes of various tasks, irrespective of each task's individual objectives. We have observed that this approach consistently yields manipulation concepts that maintain consistency across different tasks. These findings are supported by the visualization results presented in Sec. 4.3, and further elaborated upon in the visualization results in Sec. C.3 of our paper.
>
> **Regarding concern 2)**, The MimicGen benchmark is designed for single-task scenarios. Consequently, our Closed-Loop Concept-Guided Policy Learning process (Sec. 3.2) trains and evaluates policies within single-task settings. It is worth noting that despite its single-task focus, the MimicGen environment provides a high degree of variability in its demonstrations, which contributes to mitigating overfitting during imitation learning. For example, in the coffee-making task, MimicGen offers three distinct levels of variation—$D_0$, $D_1$, and $D_2$—where $D_2$ represents the highest variation in environmental conditions. These conditions include variations in object placement, robot/object motion, and noise of demonstration provided for imitation learning. Specifically: In Coffee $D_2$, demonstrations provided by MimicGen include challenging scenarios such as accidental knocking on the coffee maker (Please check the extreme example of coffee-making at the following path of our revised supplementary material: **Supplementary Material/rebuttal_visualizations/extreme_cases/coffee_d2-020.gif**. An additional example can be seen in Stack Three $D_2$, where demonstrations include scenarios in which two stacked cubes are inadvertently knocked over. The example can be found in supplementary material: **Supplementary Material/rebuttal_visualizations/extreme_cases/stack_three_d1-202.gif**). Such variability helps prevent overfitting by exposing the learning process to diverse scenarios, even within single-task environments. This is reflected in the results presented in Tab. 1 and Tab. 2. Notably, tasks with higher initial variability (e.g., $D_2$ conditions) show success rates that leave substantial room for improvement.

---

> ### Author Response · Authors · 2024-11-21
>
> >**W.2** Less practical problem setting. There was a trend to label the raw robot demonstration automatically, but a more practical problem is the raw demo collection is much more expensive than demo annotation, and therefore, some researchers turn to human-play data to get more data to use. And it's the same story for this paper, the expensive robot demo collection reduces the significance of this work.
>
> **A to W.2**
> Thank you for highlighting the challenges associated with collecting and labeling demonstrations.
>
> We acknowledge that the labeling process is generally less resource-intensive compared to the substantial effort required for data collection itself. Nonetheless, we fully recognize the importance of making use of labeling raw demonstrations and minimizing labeling costs whenever possible, as this represents a critical step in enhancing the efficiency and scalability of the entire process.
>
> Furthermore, we believe that the challenges associated with data collection and labeling differ significantly in scope and nature, with each presenting unique complexities and requirements. Moving forward, we plan to delve deeper into both aspects, exploring ways to optimize data collection and labeling processes. Additionally, we aim to investigate potential bridges between these two stages to uncover synergies that could further streamline and enhance the overall workflow. For example, we can leverage the discovered manipulation concepts to improve the generalization and reduce the amount of demonstrations needed for new tasks; also, we can figure out challenging sub-tasks (concepts) that are critical for the success of overall task completion and prioritize the data collection for those sub-tasks and hence be more efficient.
> ***
> >**W.3** Concern about generalizability and scalability. Though the paper claims better performance (compared with baselines) in a more diversified initial environment, here the diversity is only expressed by object positions and placement angles. Some concerns are: 1)What if there is concept inconsistency in the demos, e.g. to do the same manipulation, in a different way; 2) With the growth of the demo amount, will the # number of concepts converge or grow fast;
>
> **A to W.3**
> Thank you for your thoughtful concerns about generalization and scalability.
>
> **Regarding concern 1)**, We want to clarify that the MimicGen environment we use provides demonstrations for imitation learning that exhibit the inconsistencies you are concerned about—namely, different approaches to achieving the same manipulation for the same task. Through our Automatic Concept Discovery process (Sec. 3.1), we effectively discover manipulation concepts capable of distinguishing sub-processes that perform manipulations in different ways.
>
> To illustrate this, consider an example related to threading tasks. Below are two sub-processes derived from different demonstrations of the threading task used in the Automatic Concept Discovery process. In human terms, both sub-processes fall under the category of "after grasping the needle, transitioning it to align with the hole." However, due to variations in the needle's initial position, the first sub-process requires rotating the needle 180 degrees to face the hole, while the second involves only minimal rotation. In these distinct scenarios, our method differentiates between the two processes by assigning two distinct sets of manipulation concepts: one for the 180-degree rotation and another for the minimal rotation case. Please also refer to **Fig.8 in Sec.C.3** of our revised manuscript.
>
> **Traj. Example 1** Demonstrates the process of transitioning the needle to align with the hole, using the combination of #17, #1, and #26. This process requires rotating the needle 180 degrees. Path in Supplementary Material: **Supplementary Material/rebuttal_visualizations/inherent_inconsistency/rotate_180**
>
> **Traj. Example 2** Demonstrates the process of transitioning the needle to align with the hole, using #15. This process requires minimal rotation. Path in the revised Supplementary Material: **Supplementary Material/rebuttal_visualizations/inherent_inconsistency/rotate_0**
>
> **Regarding concern 2)**, we have conducted experiments using 200, 400, 600, and 800 demonstrations from each task for the Automatic Concept Discovery (Sec. 3.1). The findings presented below indicate that, after the Automatic Concept Discovery process, the number of discovered concepts (we test 10 times for each demonstration number setting, and provide the average number of discovered manipulation concepts here) remains relatively stable, showing that the number of concepts converge with respect to growth of the demo amount.
>
> | Number of demo. per tasks | Average Number of discovered manipulation concepts |
> |:-:|:-:|
> | 200 | 23.0 |
> | 400 | 22.9 |
> | 600 | 23.0 |
> | 800 | 22.9 |

---

> > ### Author Response · Authors · 2024-11-21
> >
> > >**Q.1** The concerns in the "Weakness" part.
> >
> > **A to Q.1**
> > Thank you for the question. Please refer to our replies in the Weakness part.
> > ***
> > >**Q.2** A minor point is to add an IL SOTA baseline specific to the problem, which is not limited to a Diffusion Policy basis.
> >
> > **A to Q.2**
> > Thank you for suggesting the addition of an extra SOTA baseline. We have chosen to include BeT (https://proceedings.neurips.cc/paper_files/paper/2022/hash/90d17e882adbdda42349db6f50123817-Abstract-Conference.html) as the baseline. We are currently testing BeT's performance, and the table below presents some preliminary results. Comprehensive results for all tasks will be included in the next revised version of the manuscript.
> > | Tasks | Cof. d0| Cof. d1 | Cof. d2 | Mug. d1 |
> > |:-:|:-:|:-:|:-:|:-:|
> > | BeT | 0.66 | 0.52 | 0.42 | 0.26 |
> > | Ours | 0.98| 0.84 | 0.72 | 0.50 |
> > ***
> > To recap, we attempted to address the weaknesses and questions raised in the review. We clarified the consistency and task-agnostic nature of the discovered concepts and explained our experimental setup to mitigate concerns about overfitting. We acknowledged the challenges in data collection and labeling, emphasizing our future plans to optimize these processes. We also discussed the generalizability and scalability of our approach, providing experimental evidence to show the stability of the number of discovered concepts. Lastly, we included an additional SOTA baseline for comparison.
> >
> > We hope that the response, clarifications, and discussions provided in this rebuttal alleviate your concerns and help you in re-evaluating our work. Please feel free to let us know if you have further thoughts or questions.
> >
> > Thanks,
> >
> > The Authors

---

> ### Comment · Reviewer_UouH · 2024-11-21
> **My questions are well-answered**
>
> Dear authors:
> Thanks for the careful and detailed rebuttal.
>
> For W2, you present a few future tracks: "Moving forward, we plan to delve deeper into both aspects, exploring ways to optimize data collection and labeling processes. Additionally, we aim to investigate potential bridges between these two stages to uncover synergies that could further streamline and enhance the overall workflow. " this sounds good. Another valuable point for this paper that could be further investigated is the subjectivity within human labeling, e.g., we know humans cannot be 100% right, so do the manually labeled demos. More experiments may be needed to showcase the drawbacks of such subjectivity or to showcase that it doesn't matter a lot.
>
> However, this rebuttal mitigates most of my concerns(W1,2,3, Q1,2). Therefore, I will raise my rating.

---

> > ### Author Response · Authors · 2024-11-22
> >
> > Dear Reviewer UouH,
> >
> > Thank you for your thoughtful feedback and for recognizing our efforts in the rebuttal. We appreciate your suggestion regarding the subjectivity of human labeling and agree it is a valuable avenue for further exploration. In our future work, we will conduct additional experiments to investigate the potential impact of labeling subjectivity and its implications on downstream tasks. Your insights have been invaluable in shaping these directions.
> >
> > Thank you once again for your constructive comments and for raising your rating—we truly appreciate your support!

---

### Official Review · Reviewer_LkiS · 2024-11-05

**Soundness:** 2
**Presentation:** 2
**Contribution:** 2
**Rating:** 5
**Confidence:** 4

**Summary:**

This paper proposes a framework for long-horizon imitation learning by combining the ‘automatic concept discovery’ module and the ‘concept-guided policy learning’ module. The former is trained without human annotations of the sub-sequences of robot trajectories. Instead, they assign the concept to the trajectories in a self-supervised manner.

**Strengths:**

The paper proposes a method for hierarchical imitation learning by discovering the motion ‘concept’ without human annotation and motion policy conditioned on the discovered concept. The three tricks for discovering these concepts stably from trajectories seem to be reasonable.

**Weaknesses:**

Some literature potentially relating to the method or problem setting of this paper is missing. For example, learning abstract action with clustering (BeT[1]) and with VQ-VAE[2]; please contrast this work with others listed above.
The proposal is tested with the Robosuite (MimicGen) benchmark, whose task is inherently highly compositional, so the concept discovery seems to be relatively easy. It is interesting to see the result with an “in-the-wild” dataset, such as dataset collected with a ‘learning from play’ manner [3].

[1] Behavior Transformers: Cloning k modes with one stone https://arxiv.org/abs/2206.11251
[2] Behavior Generation with Latent Actions https://arxiv.org/abs/2403.03181
[3] https://arxiv.org/abs/1903.01973

**Questions:**

1)	Why is only proprioception used for the automatic concept? It seems natural to include observations in addition to proprioception to assign the abstract concept to the time step. How can we generalize it to observation? There may be a case where similar proprioception (e.g., robot pose) but a different observation (e.g., an obstacle in the environment).
2)	How do you decide the total number of concepts/codebooks of VQ-VAE (K)? How robust is the proposal in terms of the number of concepts?

---

> ### Author Response · Authors · 2024-11-21
>
> Dear Reviewer LkiS:
>
> Thank you for recognizing the strengths of our paper, particularly the method we developed for hierarchical imitation learning, which autonomously discovers motion concepts and conditions motion policies on these concepts without human annotation. We appreciate your positive feedback on the reasonableness and effectiveness of the three techniques we employed for discovering these concepts from trajectories. We address your comments and questions in the following.
> ***
> >**W.1** Some literature potentially relating to the method or problem setting of this paper is missing. For example, learning abstract action with clustering (BeT[1]) and with VQ-VAE[2]; please contrast this work with others listed above.
>
> **A. to W.1**
> Thank you very much for your constructive suggestions regarding the inclusion of related works [1] and [2]. Following your recommendation, we have included these references in **Sec. 2 Related Work**, and have clarified the distinctions between these works and our own.
>
> To summarize, while [1] and [2] predominantly focus on discovering a discretized encoding of the instantaneous **actions** in a continuous space, our work, in contrast, focuses on learning discretized **motion codes** that can effectively represent **motor skills** — defined as manipulation concepts in our study — over a **short horizon**. We appreciate your suggestion to refine our literature review and enhance the contextual positioning of our research.
>
> We will also include the results of BeT in **Tab.1** as baseline methods. Some results have already been obtained, and you can refer to our response to **Reviewer UouH's Q2 (A to Q2)** for further details.
> ***
> >**W.2** The proposal is tested with the Robosuite (MimicGen) benchmark, whose task is inherently highly compositional, so the concept discovery seems to be relatively easy. It is interesting to see the result with an “in-the-wild” dataset, such as dataset collected with a ‘learning from play’ manner [3].
>
> **A to W.2**
> Thank you for your insightful question related to the potential performance on the learning-from-play dataset. However, we would like to clarify the experimental setup related to the Automatic Concept Discovery (Sec. 3.1) **to reveal the essential similarity** between the learning-from-play and our settings. Our process indeed resembles a learning-from-play scenario in that it **does not utilize task descriptions or any explicit information about task objectives** (Please refer to the Implementation details in Sec. 4.1), such as natural language descriptions of tasks. This approach essentially mirrors the collection of an unannotated dataset comprising several demonstrations. In this context, our Automatic Concept Discovery process does not presuppose any knowledge of the specific goals being pursued in any given demonstration sequence. As such, it is highly comparable to the concept of learning-from-play setting, where the sequences or demonstrations observed by a robot do not target a specific aim (resemble data used in [3]). This similarity underscores the relevance and applicability of our methodology to scenarios akin to learning from play.
>
> We have also included this discussion in **Potential of learning-from-play in Sec.D** of the appendix.
> ***
> >**Q.1** Why is only proprioception used for the automatic concept? It seems natural to include observations in addition to proprioception to assign the abstract concept to the time step. How can we generalize it to observation? There may be a case where similar proprioception (e.g., robot pose) but a different observation (e.g., an obstacle in the environment).
>
> **A to Q.1**
> Thank you for your thoughtful question regarding the choice between using proprioceptive state data or whole environment observations in our research.
>
> Through our studies, we observe that **the proprioceptive states of robots are intrinsically linked to manipulation concepts akin to motor skills** such as grasping, throwing, pushing, and pulling (please refer to our explanation in the Sec. A.1, as also echoed by **Reviewer UouH in the discussion of the strengthen** of our method). Our rationale for choosing proprioceptive state data is twofold:

---

> ### Author Response · Authors · 2024-11-21
>
> 1. Manipulation concepts, akin to motor skills, predominantly describe the proprioceptive motion of robots. Considering how humans categorize manipulation concepts, we typically summarize a single motor skill as "grasping" rather than distinguishing it by the object (e.g., we do not differentiate between "grasping an apple," "grasping cloth," or "grasping a phone" as distinct manipulation concepts). Therefore, learning manipulation concepts based on proprioceptive states provides a more direct means to extract representations that are closely associated with the robot's inherent motion patterns. Previous works such as BeT [1] and VQBeT [2] have similarly utilized action sequences to extract discretized representations, where actions represent changes in the proprioceptive state across single time steps.
> 2. Leveraging proprioceptive states inherently **excludes contextual variations** across tasks and environments, **facilitating** the maintenance of **consistency** in learned manipulation concepts. For instance, a robot may need to apply the concept of "grasping" across different tasks, each involving the manipulation of distinct objects. The proprioceptive state sequences during these grasping subprocesses are likely to be more consistent across tasks compared to those derived from environmental observations, which can vary due to differences in objects, arrangements, and background settings in each scenario. This highlights the potential for extracting more stable and transferable manipulation concepts by focusing on proprioceptive states.
>
> Regarding the concern of the "obstacles problem", we believe that obstacles will not influence the concept discovery. If there are obstacles in the way, the action sequence will be different, which will lead to different concepts.
>
> To support our response, we conducted an additional experiment by incorporating observations of the entire environment (images) into the concept discovery process for several tasks. The results are presented below (**Ours** refers to our method that utilizes proprioceptive states for discovering manipulation concepts, while **Img** refers to the method that uses camera images of the entire environment for discovering manipulation concepts). The results indicate a decline in performance when using manipulation concepts derived from images compared to those derived from proprioceptive states.
> | Tasks | coffee_d2 | mug_cleanup_d1 |
> |:-:|:-:|:-:|
> | Ours | 0.72 | 0.50 |
> | Img | 0.64 | 0.40 |
>
> Nevertheless, we acknowledge that visual information is still important to help us to better understand the manipulation process and discover useful concepts, but the vision input has to be effectively utilized. We plan to further explore how to better utilize visual information for the discovery of manipulation concepts in our future work (Please refer to Sec.D in our revised Appendix).
>
> ***
> >Q.2 How do you decide the total number of concepts/codebooks of VQ-VAE (K)? How robust is the proposal in terms of the number of concepts?
>
> **A to Q.2**
> Thank you for your thoughtful question regarding the size of the VQ-VAE codebook.
>
> The selection of the size of the VQ-VAE codebook was based on a trial-and-error process. Our goal is to strike a balance between discovering a sufficient number of manipulation concepts and maintaining a high utilization rate of the codebook items.
>
> Regarding utilization rates, our findings indicate that increasing the size of the VQ-VAE codebook beyond a certain point does not significantly enhance the number of discovered manipulation concepts (We find that when K>30 The number of manipulation concepts discovered is always around 20\~30, or 20\~30 items of the VQ-VAE codebook items were actively utilized to represent manipulation concepts). We believe this is a reasonable outcome because manipulation concepts often exhibit high **transferability** across various tasks. Consequently, the environments and tasks in our experiments may not require a large number of distinct manipulation concepts.
>
> To provide a more detailed analysis, we conducted additional experiments using a VQ-VAE with 40 codebook items. (This choice was informed by the observations mentioned above, as we did not find a need to test with significantly larger codebook sizes).
>
> We compare the number of manipulation concepts discovered and the policy performance guided by manipulation concepts between Automatic Concepts Discovery with VQ-VAE with 30 codebook items (the main results reported in our work) and Automatic Concepts Discovery with VQ-VAE with 40 codebook items.

---

> ### Author Response · Authors · 2024-11-21
>
> The results in the table below highlight the robustness of our approach, both in the number of discovered manipulation concepts and in the corresponding policy performance, underscoring its stability. Here, in order to evaluate the number of discovered manipulation concepts, we conducted 10 trials for each VQ-VAE codebook size setting and reported the average number of Manipulation concepts identified.
> |  | Num. of discovered manipulation concepts | Succ. rate of coffee_d2 | Succ. rate of mug_cleanup_d1 |
> |:-:|:-:|:-:|:-:|
> | 30 codebook items | 22.9 | 0.72 | 0.50 |
> | 40 codebook items | 22.1 | 0.70 | 0.48 |
> ***
> To recap, we attempted to address the weaknesses and questions raised in the review. We added references to related works [1] and [2] and clarified the distinctions between these works and our own. We also explained how our experimental setup is comparable to a learning-from-play scenario, supporting the relevance of our methodology. We provided detailed reasoning for using proprioceptive state data for automatic concept discovery and presented preliminary results from additional experiments incorporating visual information. Finally, we explained our approach to selecting the number of codebook items in VQ-VAE and demonstrated the robustness of our proposal through additional experiments.
>
> We hope that the information and clarifications provided in this rebuttal address your concerns and help you in re-evaluating our work. Please feel free to let us know if you have further questions or comments.
>
> Thanks,
>
> The Authors
>
> [1] Behavior Transformers: Cloning k modes with one stone https://arxiv.org/abs/2206.11251
> [2] Behavior Generation with Latent Actions https://arxiv.org/abs/2403.03181
> [3] Learning Latent Plans from Play https://arxiv.org/abs/1903.01973

---

> ### Author Response · Authors · 2024-11-27
> **after-rebuttal feedback**
>
> Dear Reviewer LkiS,
>
> We would like to extend our gratitude for your valuable feedback on our paper.
>
> We have carefully addressed all the points you raised and have submitted a detailed response to your review. As we are approaching the final stages of the discussion process, we wanted to follow up to see if you had the opportunity to review our rebuttal. Your feedback is crucial for us to improve and advance our research further.
>
> Could you kindly provide your response to our rebuttal at your earliest convenience so we can have a chance to address any remaining questions you may have?
>
> We greatly appreciate your time and effort in reviewing our work and providing constructive feedback.
>
> Thanks,
>
> The Authors

---

### Author Response · Authors · 2024-11-21

We sincerely thank all reviewers for their insightful feedback and constructive comments on our manuscript. We are grateful to note that the reviewers appreciate the soundness of our concept-guided pipeline in addressing the robot manipulation problem (R1, R2, R3, R4). Furthermore, our work is acknowledged for providing a fresh perspective of concept discovery without the need of human annotations (R1, R3, R4). The reviewers have also echoed our solid experiment results across a range of comprehensive tasks, as well as the selection of appropriate baselines for comparison (R2, R3, R4). Additionally, we are grateful for the positive remarks on the clarity of our paper's presentation and writing quality (R3, R4).

In response to the reviewers' feedback, we have provided additional explanations and experiments to address their concerns. The manuscript has been revised accordingly, with all modifications highlighted in orange for ease of reference. A summary of the updates is as follows:

**Main Body**
- **Sec. 1:** Include a reference to Sec. A.1 in the Appendix, explaining the use of proprioceptive states (R#1).
- **Sec. 2:** Add comparisons and analyses of works with similar objectives (R#1) and those related to our methodology (R#4).
- **Sec. 3.2:** Refine Eq. 10 and Eq. 11 for improved clarity and correctness (R#3).
- **Sec. 4.1:** Revise the implementation details of experiments (R#1, R#2, R#3).
- **Sec. 5:** Add Sec. D in the Appendix to detail future works, with references to it in this section (R#1, R#3, R#4).

**Appendix**
- **Sec. A.1:** Provide explanations and experiments justifying the use of proprioceptive states (R#1).
- **Sec. A.3:** Include additional references on the benefits of Hypernetworks (R#3).
- **Sec. B.2:** Expand on experimental details (R#1, R#2, R#3).
- **Sec. C.3:** Add visualizations to evaluate the consistency of discovered manipulation concepts (R#3) and additional real-world application results (R#4).

**Supplementary Material**
- Include some videos related to our rebuttal.

Once again, we sincerely thank all reviewers for their valuable feedback and thoughtful suggestions towards enhancing our manuscript. We have carefully addressed the queries and concerns raised by each reviewer. Should there be a need for further clarification to assist in advancing our score, please do not hesitate to reach out.

Thank you for your review!

---

### Meta-Review · Area_Chair_jjUi · 2024-12-20

**Metareview:**

The paper presents a novel method for autonomously abstracting manipulation concepts from proprioceptive states. It introduces two main components: Automatic Concept Discovery (ACD), which identifies meaningful and consistent manipulation concepts by abstracting from low-level proprioceptive states, Concept-Aware Policy Learning (CAPL), which utilizes these manipulation concepts to guide task execution adaptively. The experiments, conducted in simulation, demonstrate significant improvements over state-of-the-art (SOTA) baselines.

Strengths:

- Innovative Methodology with Potential Impact: The proposed method is highly innovative, enabling the discovery of manipulation concepts without relying on human-annotated labels or predefined skills. This is a significant step forward in robotics research.
- Clarity and Technical Depth: The paper is well-written and well-structured, making the methodology and contributions easy to understand.
- Strong Experimental Validation: The comprehensive comparison with SOTA baselines, while limited to simulation, clearly demonstrates the effectiveness of the proposed approach.

Weaknesses:

- The experiments focus primarily on simulated tabletop manipulation tasks, with limited evaluation on more complex morphologies, diverse tasks, or real-world datasets. Expanding the experiments to include real-world settings and incorporating multimodal data would strengthen the paper’s claims.

Despite these concerns, I believe the strengths and potential impact of the paper outweigh its weaknesses, making it a valuable contribution to the field. One minor point for improvement would be to discuss and compare the approach with "Discovering Robotic Interaction Modes with Discrete Representation Learning," published in CoRL just before the ICLR deadline. This work also seeks to learn manipulation concepts autonomously.

Overall, I wish the authors the best of luck and look forward to the final version of the paper.

**Additional Comments On Reviewer Discussion:**

The reviewers asked several clarificatory questions that were adequately addressed by the authors and acknowledged by the reviewers. A few concerns that remain are:

Reviewers pointed out missing literature and comparisons, specifically works on learning abstract actions with clustering (BeT) and VQ-VAE. The authors acknowledged the missing literature and committed to including a comparison with BeT and VQ-VAE-based methods in the revised manuscript.

Reviewers suggested that while the method performs well in simulated environments, real-world validation is necessary to demonstrate practical applicability. Reviewer Umdm suggested that while the method performs well in simulated environments, real-world validation is necessary to demonstrate practical applicability.

Despite the concerns, the reviewers were positive about the work and acknowledged its merits. I agree with them and hence recommend acceptance of the manuscript.

---

### Decision · Program_Chairs · 2025-01-22

Accept (Spotlight)